



# 1  The Canadian Earth System Model version 5 (CanESM5.0.3)

Neil C. Swart[1,3], Jason N.S. Cole[1], Viatcheslav V. Kharin[1], Mike Lazare[1], John F. Scinocca[1], Nathan P.
Gillett[1], James Anstey[1], Vivek Arora[1], James R. Christian[1,2], Sarah Hanna[1], Yanjun Jiao[1], Warren G. Lee[1],
Fouad Majaess[1], Oleg A. Saenko[1], Christian Seiler[4], Clint Seinen[1], Andrew Shao[3], Larry Solheim[1], Knut
von Salzen[1,3], Duo Yang[1], Barbara Winter[1]
[1]Canadian Centre for Climate Modelling and Analysis, Environment and Climate Change Canada, Victoria, BC, V8W 2P2,
Canada
[2]Fisheries and Oceans Canada, Institute of Ocean Sciences, Sidney, BC, Canada.
[3]University of Victoria, 3800 Finnerty Rd, Victoria, BC, V8P 5C2, Canada.
[4]Climate Processes Section, Environment and Climate Change Canada, Victoria, BC, V8P 5C2, Canada.
*Correspondence to*: Neil C. Swart (neil.swart@canada.ca)
**Abstract.** The Canadian Earth System Model version 5 (CanESM5) is a global model developed to simulate historical climate
change and variability, to make centennial scale projections of future climate, and to produce initialized seasonal and decadal
predictions. This paper describes the model components and their coupling, as well as various aspects of model development,
including tuning, optimization and a reproducibility strategy. We also document the stability of the model using a long control
simulation, quantify the model's ability to reproduce large scale features of the historical climate, and evaluate the response of
the model to external forcing. CanESM5 is comprised of three dimensional atmosphere (T63 spectral resolution / 2.8°) and
ocean (nominally 1°) general circulation models, a sea ice model, a land surface scheme, and explicit land and ocean carbon
cycle models. The model features relatively coarse resolution and high throughput, which facilitates the production of large
ensembles. CanESM5 has a notably higher equilibrium climate sensitivity (5.7 K) than its predecessor CanESM2 (3.8 K),
which we briefly discuss, along with simulated changes over the historical period. CanESM5 simulations are contributing to
the Coupled Model Intercomparison Project Phase 6 (CMIP6), and will be employed for climate science and service
applications in Canada.



## 1 Introduction

A multitude of evidence shows that human influence is driving accelerating changes in the climate system, which are unprecedented in millennia (IPCC, 2013). As the impacts of climate change are increasingly being felt, so is the urgency to take action based on reliable scientific information (UNFCCC, 2015). To this end, the Canadian Centre for Climate Modelling and Analysis (CCCma) is engaged in an ongoing effort to improve modelling of the global earth system, with the aim of enhancing our understanding of climate system function, variability and historical changes, and for making improved quantitative predictions and projections of future climate. The global coupled model, CanESM, forms the basis of the CCCma modelling system, which also includes the Canadian Regional Climate Model (CanRCM) for finer scale modelling of the atmosphere (Scinocca et al., 2016), the Canadian Middle Atmosphere Model (CMAM) with atmospheric chemistry (Scinocca et al., 2008), and the Canadian Seasonal to Interseasonal Prediction System which is used for seasonal prediction and decadal forecasts (CanSIPS, Merryfield et al., 2013).

CanESM5 is the current version of CCCma's global model, and has a pedigree extending back 40 years to the introduction of the first atmospheric General Circulation Model (GCM) developed at CCCma's predecessor, the Canadian Climate Centre (Boer and McFarlane, 1979; Boer et al., 1984; McFarlane, et al., 1992). Successive versions of the model introduced a dynamic three dimensional ocean in CGCM1 (Flato et al., 2000; Boer et al. 2000a; Boer et al. 2000b), and later an interactive carbon cycle was included to form CanESM1 (Arora et al, 2009; Christian et al., 2010). The last major iteration of the model, CanESM2 (Arora et al, 2011), was used in the Coupled Model Intercomparison Project phase 5 (CMIP5), and continues to be employed for novel science applications such as generating large initial condition ensembles for detection and attribution (e.g. Kirchmeier-Young et al., 2017; Swart et al., 2018).

As detailed below, CanESM5 represents a major update to CanESM2. The update includes incremental improvements to the atmosphere, land surface and terrestrial ecosystem models. The major changes relative to CanESM2 are the implementation of completely new models for the ocean, sea-ice, marine ecosystems, and a new coupler. Model developers have a choice in distributing increasing, but finite, computational resources between improvements in model resolution, model complexity and model throughput (i.e. number of years simulated). The resolution of CanESM5 (T63 or ~2.8° in the atmosphere and ~1° in the ocean) remains similar to CanESM2, and is at the lower end of the spectrum of CMIP6 models. The advantage of this coarse resolution is a relatively high model throughput given the complexity of the model, which enables many years of simulation to be achieved with available computational resources. The first major application of CanESM5 is CMIP6 (Eyring et al., 2016), and over 50,000 years of simulation are being conducted for the 20 CMIP6-endorsed MIPs in which CCCma is participating.





The aim of this paper is to provide a comprehensive reference that documents CanESM5. In the sections below, each of the
model components is briefly described, and we also explain the approach used to develop, tune and numerically optimize the
model. Following that, we document the stability of the model in a long pre-industrial control simulation, and the model's
ability to reproduce large-scale features of the climate system. Finally, we investigate the sensitivity of the model to external
forcings.
**2 Component Models**
In CanESM5 the atmosphere is represented by the Canadian Atmosphere Model (CanAM5), which incorporates the Canadian
Land Surface Scheme (CLASS) and the Canadian Terrestrial Ecosystem Model (CTEM). The ocean is represented by a
CCCma customized version of the Nucleus for European Modelling of the Ocean model (NEMO), with ocean biogeochemistry
represented by either the Canadian Model of Ocean Carbon (CMOC) in the standard model version labelled as *CanESM5*, or
the Canadian Ocean Ecosystem model (CanOE) in versions labelled *CanESM5-CanOE*. The atmosphere and ocean
components are coupled by means of the Canadian Coupler (CanCPL). These components of CanESM5 are summarized
schematically in Fig. 1, and described further below.

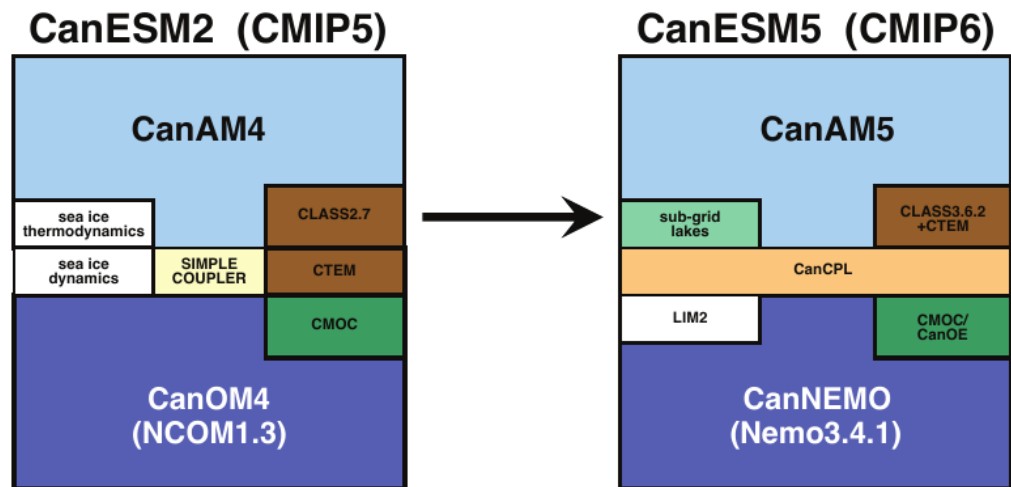

**Figure 1: Model schematic, showing the evolution of components between CanESM2 and CanESM5.**

**2.1 The Canadian Atmospheric Model version 5 (CanAM5)**
Version 5 of the Canadian Atmospheric Model (CanAM5) has several improvements relative to its predecessor, CanAM4 (von
Salzen et al., 2013), including changes to aerosol, clouds, radiation, land surface and lake processes. The model uses a T63





triangular spectral truncation in the model dynamical core, with an approximate horizontal resolution of 2.8 degrees in
latitude/longitude. It uses a hybrid vertical coordinate system with 49 levels between the surface and 1 hPa, with a vertical
resolution of about 100 m near the surface. Relative to the 35 levels used in CanESM2 most of the additional 14 levels were
added in the upper troposphere and stratosphere. The representation of radiative processes was improved through changes to
the parameterization of albedos for bare soil, snow and ocean white-caps; cloud optics for ice clouds and polluted liquid clouds;
improved aerosol optical properties; and absorption by the water vapour continuum at solar wavelengths. For aerosols, the
emission of mineral dust and dimethyl sulfide (DMS) was improved while for clouds a parameterization of the second indirect
effect          was          activated          in          the          stratiform          cloud          microphysics.

Parameterizations of surface processes were improved through an upgrade of the land surface scheme from CLASS 2.7 to
3.6.2 as well as the inclusion of a parameterization for sub-grid lakes. CanESM5 represents the first coupled model produced
by the CCCma in which the atmosphere and ocean do not employ coincident horizontal computational grids. As a
consequence, CanAM5 was modified to support a fractional land mask, by generalizing its underlying surface to support grid-
box fractional tiles of land and water. This tiling technology was extended to include land surface components of ocean, sea-
ice and subgrid scale lakes. In this way appropriate fluxes are provided to each model component. A more detailed description
of CanAM5 will be provided in a companion paper in this special issue (Cole et al., 2019).
**2.2 CLASS-CTEM**
The CLASS-CTEM modelling framework consists of the Canadian Land Surface Scheme (CLASS) and the Canadian
Terrestrial Ecosystem Model (CTEM) which together form the land component of CanESM5. CLASS and CTEM simulate
the physical and biogeochemical land surface processes, respectively, and together they calculate fluxes of energy, water, $CO_2$
and wetland $CH_4$ emissions at the land-atmosphere boundary. The introduction of dynamic wetlands and their methane
emissions is a new biogeochemical process added since the CanESM2.

CLASS is described in detail in Verseghy (1991), Verseghy et al. (1993) and Verseghy (2000) and version 3.6.2 is used in
CanESM5. It prognostically calculates the temperature for its soil layers, their liquid and frozen moisture contents, temperature
of a single vegetation canopy layer if it is present as dictated by the specified land cover, and the snow water equivalent and
temperature of a single snow layer if it is present. Three permeable soil layers are used with default thicknesses of 0.1, 0.25
and 3.75 m. The depth to bedrock is specified on the basis of the global data set of Zobler (1986) which reduces the thicknesses
of the permeable soil layers. CLASS performs energy and water balance calculations and all physical land surface processes
for four plant functional types (PFTs) (needleleaf trees, broadleaf trees, crops and grasses), and operates at the same sub-daily
time step as the rest of the atmospheric component.





CTEM models photosynthesis, autotrophic respiration from its three living vegetation components (leaves, stem and roots)
and heterotrophic respiration fluxes from its two dead carbon components (litter and soil carbon) and is described in detail in
Arora (2003), Arora and Boer (2003) and Arora and Boer (2005). Land use change is also modelled on the basis of specified
time-varying land cover which incorporates the increase in crop area over the historical period following Arora and Boer
(2010). CTEM's photosynthesis module operates within CLASS, at the same time step as rest of the atmospheric component.
CTEM provides CLASS with dynamically simulated structural attributes of vegetation including leaf area index (LAI),
vegetation height, rooting depth and distribution, and above ground canopy mass. All terrestrial ecosystem processes other
than photosynthesis are modelled in CTEM at a daily time step. Terrestrial ecosystem processes in CTEM are modelled for
nine PFTs that map directly to the PFTs used by CLASS. Needleleaf trees are divided into their deciduous and evergreen types,
broadleaf trees are divided into cold and drought deciduous and evergreen types, and crops and grasses are divided into $C_3$ and
$C_4$ versions based on their photosynthetic pathways.

The calculation of wetland extent and methane emissions from wetlands is described in detail in Arora et al. (2018). In brief,
dynamic wetland extent is based on the "flat" fraction in each grid cell with slopes less than 0.2%. As the liquid soil moisture
in the top soil layer increases above a specified threshold, the wetland fraction increases linearly up to a maximum value, equal
to the flat fraction in a grid cell. The simulated $CH_4$ emissions from wetlands are calculated by scaling the heterotrophic
respiration flux from the model's litter and soil carbon pools to account for the ratio of wetland to upland heterotrophic
respiratory flux and the fact that some of the $CH_4$ flux is oxidized in the soil column before reaching the atmosphere.

Specified land cover that includes fractional coverages of CTEM's nine PFTs is generated based on a potential vegetation
cover for 1850 upon which the 1850 crop cover is superimposed. From 1850 onwards, as the fractional area of $C_3$ and $C_4$ crops
changes the fractional coverages of the other non-crop PFTs are adjusted linearly in proportion to their existing coverage, as
described in Arora and Boer (2010). The increase in crop area over the historical period is based on LUH2 v2h product
(http://luh.umd.edu/data.shtml) of the land use harmonization (LUH) effort (Hurtt et al., 2011).

Surface runoff and baseflow simulated by CLASS are routed through river networks. Major river basins are discretized at the
resolution of the model and river routing is performed at the model resolution using the variable velocity river routing scheme
presented in Arora and Boer (1999). The delay in routing is caused by the time taken by runoff to travel over land in an assumed
rectangular river channel and a ground water component to which baseflow contributes. Streamflow (i.e. the routed runoff)
contributes fresh water to the ocean grid cell where the land fraction of a CanAM grid cell first drops below 0.5 along the river
network as the river approaches the ocean.

In CanESM5, glacier coverage is specified and static. Grid cells are specified as glacier if the fraction of the grid cell covered
by ice exceeds 40%, based on the GLC2000 dataset (Bartholomé and Belward, 2005). The combination of this threshold and





the model resolution results in glacier covered cells predominantly representing the Antarctic and Greenland ice sheets, with
a few glacier cells in the Himalayas, Northern Canada and Alaska. Snow can accumulate on glaciers, and any additional snow
above the threshold of 100 kg m$^{-2}$ of snow water equivalent is "converted into ice", and an equivalent mass of freshwater is
immediately inserted into runoff – implicitly representing mass balance between accumulation and calving. Snow and ice on
glaciers can be melted, with the water exceeding a ponding limit inserted into runoff. There is no explicit accounting for glacier
mass balance, or adjustment of glacier coverage. This represents a potentially infinite global source or sink of fresh water in
the coupled system, particularly in climates which are far from the state represented by GLC2000. However, in practice the
timescales of our centennial-scale simulations are much shorter than the response times of ice sheet coverage, and any
imbalances are small (Section 4).
**2.3 NEMO modified for CanESM (CanNEMO)**
The ocean component is based on NEMO version 3.4.1 (Madec et al. 2012). It is configured on the tripolar ORCA1 C-grid
with 45 z-coordinate vertical levels, varying in thickness from ~6 m near the surface to ~250 m in the abyssal ocean.
Bathymetry is represented with partial cells. The horizontal resolution is based on a 1° Mercator grid, varying with the cosine
of latitude, with a refinement of the meridional grid spacing to 1/3° near the Equator. The adopted model settings include the
linear free surface formulation (see Madec et al. 2012 and references therein). Momentum and tracers are mixed vertically
using a turbulent kinetic energy scheme based on the model of Gaspar et al. (1990). The tidally-driven mixing in the abyssal
ocean is accounted for following Simmons et al. (2004). Base values of vertical diffusivity and viscosity are $0.5 \times 10^{-5}$ and
$1.5 \times 10^{-4}$ m$^2$/s, respectively. A parameterization of double diffusive mixing (Merryfield et al., 1999) is also included. Lateral
viscosity is parameterized by a horizontal Laplacian operator with eddy viscosity coefficient of $1.0 \times 10^4$ m$^2$/s in the tropics,
decreasing with latitude as the grid spacing decreases. Tracers are advected using the total variance dissipation scheme
(Zalesak, 1979). Lateral mixing of tracers (Redi 1982) is parameterized by an isoneutral Laplacian operator with eddy
diffusivity coefficient of $1. \times 10^3$ m$^2$/s at the Equator, which decreases poleward with the cosine of latitude. The process of
potential energy extraction by baroclinic instability is represented with the Gent and McWilliams (1990) scheme using a
spatially-variable formulation for the mesoscale eddy transfer coefficient, as briefly described below.

Two modifications have been introduced to the NEMO's mesoscale and small-scale mixing physics. The first modification is
motivated by the observational evidence suggesting that away from the tropics the eddy scale decreases less rapidly than does
the Rossby radius (e.g., Chelton et al., 2011). This is taken into consideration in the formulation for the eddy mixing length
scale, which is used to compute the mesoscale eddy transfer coefficient for the Gent and McWilliams (1990) scheme (for
details, see Saenko et al., 2018). The second modification is motivated by the observationally based estimates suggesting that
a fraction of the mesoscale eddy energy could get scattered into high-wavenumber internal waves, the breaking of which results
in enhanced diapycnal mixing (e.g., Marshall and Naveira Garabato, 2008; Sheen et al., 2014). A simple way to represent this





process in an ocean general circulation model was proposed in Saenko et al. (2012). Here, we employ an updated version of
their scheme which accounts better for the eddy-induced diapycnal mixing observed in the deep Southern Ocean (e.g., Sheen
et al., 2014).

CanESM5 uses the LIM2 sea ice model (Fichefet and Morales Maqueda, 1997; Bouillon et al., 2009), which is run within the
NEMO framework. Some details regarding the calculation of surface temperature over sea-ice are described in the coupling
section below.
**2.4 Ocean biogeochemistry**
Two different ocean biogeochemical models, of differing complexity and expense, were developed in the NEMO framework:
CMOC and CanOE. Two coupled models versions will be submitted to CMIP6. The version labelled as *CanESM5* uses CMOC
and was used to run all the experiments that CCCma has committed to. The version labelled *CanESM5-CanOE*, described in
another paper in this special issue (Christian et al., 2019), is identical to CanESM5, except that CMOC was replaced with
CanOE, and this version has been used to run a subset of the CMIP6 experiments, including DECK and historical (see Section
3.4). Both biogeochemical models simulate ocean carbon chemistry and abiotic chemical processes such as oxygen solubility
identically, in accordance with the OMIP-BGC protocol (Orr et al., 2017).

**2.4.1 Canadian Model of Ocean Carbon (CMOC)**
The Canadian Model of Ocean Carbon was developed for earlier versions of CanESM (Zahariev et al., 2008; Christian et al.,
2010; Arora et al., 2011), and includes carbon chemistry and biology. The biological component is a simple Nutrient-
Phytoplankton-Zooplankton-Detritus (NPZD) model, with fixed Redfield stoichiometry, and simple parameterizations of iron
limitation, nitrogen fixation, and export flux of calcium carbonate. CMOC was migrated into the NEMO modelling system,
and the following important modifications were made: i) oxygen was added as a passive tracer with no feedback on biology;
ii) carbon chemistry routines were updated to conform to the OMIP-BGC protocol (Orr et al., 2017); iii) additional passive
tracers requested by OMIP were added, including natural and abiotic DIC as well as the inert tracers CFC11, CFC12 and SF6.

**2.4.2 Canadian Ocean Ecosystem Model (CanOE)**
The Canadian Ocean Ecosystem Model (CanOE) is a new ocean biology model with a greater degree of complexity than
CMOC, and represents explicitly some processes that were highly parameterized in CMOC. CanOE has two size classes for
each of phytoplankton, zooplankton and detritus, with variable elemental (C/N/Fe) ratios in phytoplankton and fixed ratios for
zooplankton and detritus. Each detritus pool has its own distinct sinking rate. In addition, there is an explicit detrital $CaCO_3$
variable, with its own sinking rate. Iron is explicitly modelled, with a dissolved iron state variable, sources from aeolian





deposition and reducing sediments, and irreversible scavenging from the dissolved pool. $N_2$ fixation is parameterized similarly
to CMOC with temperature- and irradiance-dependence and inhibition by Dissolved Inorganic Nitrogen, but no explicit $N_2$-
fixer group. In addition, $N_2$ fixation is iron-limited in CanOE. In CanOE, denitrification is modelled prognostically and occurs
only where dissolved oxygen is <6 mmol $m^{-3}$. Deposition of organic carbon is instantaneously remineralized at the sea floor
as in CMOC, and $CaCO_3$ deposited at the sea floor dissolves if the calcite is undersaturated (whereas in CMOC the burial
fraction is implicitly 100%). Carbon chemistry and all abiotic chemical processes such as oxygen solubility conform to the
OMIP-BGC protocol (Orr et al., 2017) and are identical in CanOE and CMOC, except that in CMOC the carbon chemistry
solver is applied only in the surface layer (as there is no feedback from saturation state to other biogeochemical processes in
the subsurface layers). CanOE has roughly twice the computational expense of CMOC.
**2.5 The Canadian Coupler (CanCPL)**
CanCPL is a new coupler developed to facilitate communication between CanAM and CanNEMO. CanCPL depends on Earth
System Modeling Framework (ESMF) library routines for regridding, time advancement, and other miscellaneous
infrastructure (Theurich et al., 2016; Collins et al., 2005; Hill et al., 2004). It was designed for the Multiple Program Multiple
Data (MPMD) execution mode, with communication between the model components and the coupler via the Message Passing
Interface (MPI).

The fields passed between the model components are summarized in Tables A1 to A4. In general, CanNEMO passes
instantaneous prognostic fields, which are remapped by CanCPL and given to CanAM as lower boundary conditions. These
prognostic fields (sea surface temperature, sea-ice concentration and mass of sea-ice and snow) are held constant in CanAM
over the course of the coupling cycle. After integrating forward for a coupling cycle, CanAM passes back fluxes, averaged
over the coupling interval, which are remapped in CanCPL and passed on to NEMO as surface boundary conditions. An
exception is the ocean surface $CO_2$ flux, which is computed in CanNEMO and passed to CanAM. CanAM and CanNEMO are
run in parallel, and the timing of exchanges through the coupler is indicated schematically in Fig. 2.

All regridding in CanCPL is done using the ESMF first order conservative regridding option (ESMF, 2018), ensuring that
global integrals remain constant for all quantities passed between component models (but see an exception below). The
remapping weights $w_{ij}$, for a particular source cell $i$ and destination cell $j$ are given by: $w_{ij} = f_{ij} \times A_{si}/(A_{dj} \times D_j)$, where
$f_{ij}$ is the fraction of the source cell $i$ contributing to the destination cell $j$, $A_{si}$ and $A_{dj}$ are the areas of the source and destination
cells, and $D_j$ is the fraction of the destination cell that intersects the unmasked source grid (ESMF, 2018).


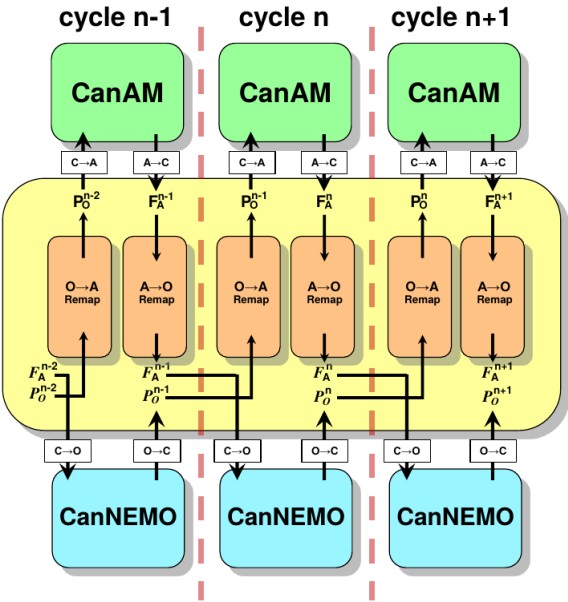

**Figure 2: Schematic showing the ordering of exchanges between CanCPL and CanAM and CanNEMO. Prognostic fields ($P_O$) are passed from NEMO to the coupler, remapped, and passed to CanAM. Fluxes ($F_A$) are passed from CanAM, remapped in CanCPL, and passed NEMO to complete the next coupling cycle. Superscripts denote the coupling cycle, e.g. Prognostic fields from NEMO are passed to CanCPL at the end of cycle "n", remapped, and used in CanAM during cycle "n+1".**

Within the NEMO coupling interface the "conservative" coupling option is employed. This option dictates that net fluxes are passed over the combined ocean-ice cell, and the fluxes over only the ice covered fraction of the cell are also supplied, in principle allowing net conservation, even if the distribution of ice has changed given the unavoidable one coupling cycle lag encountered in parallel coupling mode. It was verified that the net heat fluxes passed from CanAM were identical to the net fluxes received by NEMO, to the level of machine precision. Conservation in the coupled model piControl run is discussed further in Section 4.

Sea-ice thermodynamics are computed in the LIM2 ice model, based on the surface fluxes received from CanAM, and the basal heat flux from the NEMO liquid ocean. LIM2 provides the sea-ice concentration, snow and ice thickness to CanAM, via the coupler. The surface flux calculation in CanAM5 requires the ground temperature at the snow/sea-ice interface, $GT_{ice}$. The $GT_{ice}$ for this purpose can be passed from LIM2 to CanAM once each coupling cycle, or an alternative $GT_{ice}$ can be evaluated in CanAM at every model time step, taking into account evolving surface albedo and atmospheric temperature (e.g. West et al., 2016). As implemented, when computing $GT_{ice}$, CanAM independently computes the conductive heat flux through sea-ice, and there is no constraint that this flux, or $GT_{ice}$ is the same as that in LIM2. Conservation is maintained because the net





heat flux between the atmosphere and sea-ice is computed in CanAM and applied to LIM2, but different ice surface temperatures could result. Both approaches to computing surface fluxes were tested in CanESM5 and no major impacts on sea-ice or the broader climate system relative to the default model were discovered. However, a significantly shorter coupling cycle of one hour was required for convergence when fluxes were computed from the LIM2 $GT_{ice}$ passed through the coupler. The shorter coupling period was required to more physically resolve the response to diurnal variations in radiative and sensible heat fluxes from the atmosphere (see for example West et al., 2016). The evaluation of fluxes from $GT_{ice}$ computed in CanAM, on the other hand, was stable for coupling periods ranging from 1 to 24 hours with no major changes in the mean climate, or variability immediately apparent. A final coupling cycle interval of three hours was implemented for CanESM5 with the computation of fluxes based on the CanAM evaluation of $GT_{ice}$. These choices represented improved robustness and a compromise between greater efficiency (i.e. longer coupling periods) and maximum "realism", which would be the one hour coupling dictated by the length of the NEMO time step.

After a significant number of CMIP6 production simulations were complete, it was determined that while conservative remapping was desirable for heat and water fluxes, it introduced issues in the wind-stress field passed from CanAM to CanNEMO. Specifically, since CanAM is nominally three times coarser than CanNEMO, conservative remapping resulted in constant wind-stress fields over several NEMO grid-cells, followed by an abrupt change at the edge of the next CanAM cell. This blockiness in the wind-stress results in a non-smooth first derivative, and the resulting peaked wind-stress curl results in unphysical features in, for example, the ocean vertical velocities. Changing regridding of only wind-stresses to the more typical "bilinear" interpolation, instead of "conservative" remapping, largely alleviates this issue. Sensitivity tests indicate no major impact on gross climate change characteristics such as transient climate response or equilibrium climate sensitivity, or on general features of the surface climate. However there is an impact on local ocean dynamics, which led to the decision to submit a "perturbed" physics member to CMIP6. Hence, simulations submitted to CMIP6 labelled as perturbed physics member 1 ("p1") use conservative remapping for wind stress, while those labelled as "p2" use bilinear regridding (see Section 3.4). A comparison between p1 and p2 runs is provided in Appendix E.

## 3 Model development and deployment

### 3.1 Model tuning and spin up

Each of the CanESM5 component models, CanAM5, CLASS-CTEM and CanNEMO, were initially developed independently under driving by observations in stand-alone configurations - CanAM5 in present-day (2003-2008) AMIP mode and CanNEMO in preindustrial (PI) OMIP-like mode using CORE bulk formulae. In these configurations, free parameters were initially adjusted to reduce climatological biases assessed via a range of diagnostics. Further details of the CanAM5 tuning may be found in Cole et al. (2019). The component models were then brought together in a preindustrial configuration (i.e. the piControl experiment), which was evaluated based on an array of diagnostics. Several thousand years of coupled simulation





was run during the finalization of model, and an approach was taken whereby AMIP simulations would be used to derive parameter adjustments in CanAM, which would then be applied to the coupled model.

Initial present-day configurations of CanAM5 that were tuned to give roughly the observed top of the atmosphere net radiative forcing (TOA forcing ~0.7-1.0W m$^{-2}$) in an AMIP simulation produced coupled piControl simulations that were too cold (global mean near-surface temperatures below 12°C), with extensive sea-ice and a collapsing meridional overturning circulation. One contributor to the tendency of the new coupled model to cool was the inclusion of the thermodynamic consequences of snow melt in the open ocean, which induces an average global cooling of ~0.5 W m$^{-2}$ in the piControl, and was not included in the previous version, CanESM2.

This initial coupled-model cold bias was rectified by adjusting free parameters in CanAM, CLASS and LIM2, in order to achieve a piControl simulation with a global mean screen temperature of around 13.7°C (roughly the absolute value provided for 1850-1900 by the NASA-GISS, Berkeley Earth and HadCRUT4 datasets), and a sea-ice volume within the spread of CMIP5 models. The specific parameters adjusted were: emissivity of snow (from 1 to 0.97), snow grain size on sea-ice, the drainage parameter controlling soil moisture, the LIM2 parameter controlling the lead closure rate (from 2.0 to 3.0), and most significantly the accretion rate in cloud microphysics. The accretion rate exerted the largest control, and sensitivity to this parameter is described more fully in a companion paper (Cole et al., 2019).

The consequence of the adjustments in CanAM5 was an increase in the present day TOA forcing in AMIP mode from ~1 W/m$^2$ to ~2.5 W m$^{-2}$. Nonetheless, historical simulations of the coupled CanESM5 initialized from its equilibrated piControl show an increase in TOA forcing roughly matching the observed values of ~0.7-1.0 W m$^{-2}$ over the 2003-2008 period for which CanAM5 was tuned in AMIP mode. The difference in patterns of SST and sea-ice concentrations between the coupled model and observations are thought to be the cause of these differences in TOA balance between coupled and AMIP mode.

The final adjustment was to the carbon uptake over land so as to better match the observed value over the historical period, and achieved via the parameter which controls the strength of the $CO_2$ fertilization effect (Arora and Scinocca, 2016). No more extensive tuning of CanESM5 was undertaken. Critically, no tuning was undertaken on the climate system response to forcing - the transient and equilibrium climate sensitivity of CanESM5 are purely emergent properties. Once the tuned final configuration of CanESM5 was available, ocean potential temperature and salinity fields were initialized from World Ocean Atlas 2009, while CanAM, CLASS-CTEM and CMOC were initialized from the restarts from earlier development runs. The model was spun up for over 1500 years prior to the launch of the official CMIP6 piControl simulation, which extends for a further 2000 years.



## 3.2 Code management, version control and reproducibility

CanESM5 is the first version of the model to be publicly released, and this code sharing has been facilitated by the adoption of a new version control based strategy for code management. Additional goals of this new system are to adopt industry standard software development practises, to improve development efficiency, and to make all CanESM5 CMIP6 simulations fully repeatable.

To maintain modularity, the code is organized such that each model component has a dedicated *git* repository for the version control of its source code (Table 1). A dedicated *super repository* tracks each of the components as *git submodules*. In this way, the *super repo.* keeps track of which specific versions of each component combine together to form a functional version of CanESM. A commit of the CanESM super repo., which is representable by an 8 character truncated SHA1 checksum, hence uniquely defines a version of the full CanESM source code. The model development process follows an industry standard workflow (Table B1). New model features are merged onto the *develop_canesm* branch, which reflects the ongoing development of the model. Specific model versions, such as that used for CMIP6, are given tags and issued DOIs for ease of reference. We use an internal deployment of *gitlab* to host the model code and associated issue trackers, and we mirror the code to the public, online code hosting platform at gitlab.com/cccma/canesm.

A dedicated ecosystem of software is used to configure, compile, run, and analyze CanESM simulations on ECCC's HPC (Table B2). Several measures are taken to ensure modularity and repeatability. The source code for each run is recursively cloned from gitlab and is fully self contained. A strict checking routine ensures that any code changes are committed to the version control system, and any run-specific configuration changes are captured in a dedicated configuration repository. A database records the SHA1 checksums of the particular model version and configuration used for every run, and these are included in CMIP6 NetCDF output for traceability. Input files for model initialization and forcing are also tracked for reproducibility (Table B1).

Our strategy of version control, run isolation, strict checking and logging ensures that simulations can be repeated in the future, and the same climate will be obtained (bit identical reproducibility is a further step and is dependant on machine architecture and compilers). The implementation of a clear branching workflow, and the uptake of modern tools such as issue trackers, and the gitlab online code-hosting application has improved both collaboration and management of the code. This new system also led to large, unexpected improvements in model performance for two major reasons. The first was democratization of the code – via the promotion of group ownership of the code. The second was the freedom to experiment across the full code base ensured by our isolated run setup (Table B2), which was not possible under the previous system of using a single installed library of code shared across many runs. The performance gains achieved are described in the following section.





**Table 1: Code structure and repositories.**

| Repository | Purpose |
|---|---|
| CanESM | The top-level super-repository, which tracks specific versions of the component *submodules* listed below, to form a function version of the model. Also contains a CONFIG directory with configuration files for the model. |
| CanAM | The source code for the spectral dynamics and physics of CanAM. |
| CanDIAG | Diagnostic source code for analyzing CanAM output, this repository also contains various scripting used to run the model. |
| CanNEMO | The CCCma modified NEMO source code, along with additional utility scripting. |
| CanCPL | The coupler source code. |
| CCCma_tools | A collection of software tools for compiling, running and diagnosing CanESM on ECCC's high performance computer. |

## 3.3 Model optimization and benchmarking

The ECCC high performance computer system consists of the following components: a "backend" Cray XC40, with two 18 core Broadwell CPUs per node (for 36 cores per node), and roughly 800 nodes in total, connected to a multi-PB lustre file system used as scratch space. This machine is networked to a "frontend" Cray CS5000, with several PB of attached HPFS spinning disk. This whole compute arrangement is replicated in a separate hall for redundancy, effectively doubling the available resources. Finally, a large tape-storage system (HPNLS) is available for archiving model results.

The initial implementation of a CanESM5 precursor on this new HPC occurred around Nov 1, 2017. The original workflow roughly followed that used for CanESM2 CMIP5 simulations. All CanESM5 components (atmosphere CanAM, coupler CanCPL and ocean CanNEMO) were originally running at 64-bit precision. The atmospheric component CanAM was running on two 36-core compute nodes, the coupler was running on a separate node, and the ocean component was running on 3 nodes, resulting in 6 nodes in total. The initial throughout on the system, without queue time, was around 4.6 years of simulation per wall-clock day (ypd), or alternatively 0.02 simulation years per core-day, when normalizing by the number of cores used.

In parallel to the physical model development, significant effort was made to improve the model throughput and eliminate a number of inefficiencies in the older CMIP5 workflow (Fig. 3). The largest effort was devoted to improving the efficiency of CanAM5, since this was identified as the major bottleneck. A brief summary of the improvements is given in Table C1 and Fig. 3. The most substantial and rewarding change was in converting the 64-bit CanAM component to 32-bit numerics. Since



the remaining two components, CanCPL and CanNEMO are still running at the 64-bit precision, the communication between
CanAM and CanCPL required the promotion of a number of variables from 32-bit precision to 64-bit and back. The 32-bit
CanAM implementation required a number of modifications to maintain the numerical stability of the code. Calculations in
some subroutines, most notably in the radiation code, were promoted to the 64-bit accuracy. Conservation of some tracers, in
particular $CO_2$, was compromised at the 32-bit precision, and some additional code changes to conserve $CO_2$ and maintain
carbon budgets were implemented. Significant effort was also invested in optimizing compiler options used for NEMO to
maximize efficiency, while the scalability of the NEMO code allowed sensibly increasing the node count to keep pace with
the accelerated 32-bit version of CanAM.

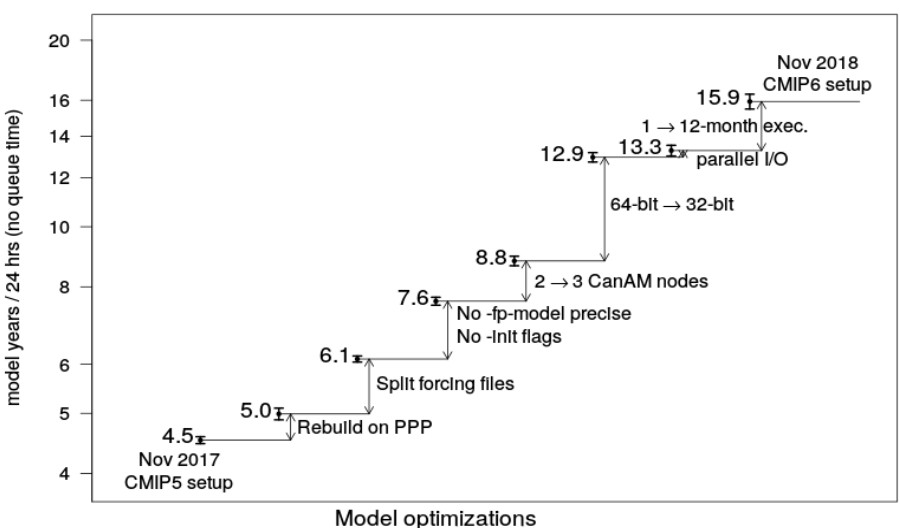


**Figure 3: Schematic of CanESM5 optimization**

In the final setup, the CanAM/CanCPL components are running on three shared compute nodes, and the ocean component
CanNEMO is running on 5 nodes, resulting in 8 nodes overall. The combined effect of the improvements listed in Table C1
resulted in more than tripling the original throughput to about 16 ypd (Fig. 3). Despite the increase in the total node count from
6 to 8, the efficiency of the model also improved roughly three fold, from 0.02 simulation years per core day of compute to
about 0.06 years per core day. This final model configuration can complete a realization of the 165 year CMIP6 historical
experiment in just over 10 days, compared to about 36 days had no optimization been undertaken. At the time of writing, over
50,000 years of CMIP6 related simulation had been conducted with CanESM5, consuming about one million core-days of
compute time, resulting in about 8 PB of data archived to tape, and over 100 TB of data publicly served on the Earth System
Grid Federation (ESGF).



## 3.4 Model experiments and scientific application

This section describes the major experiments and model variants of CanESM5 that are being conducted for the Coupled Model Intercomparison Phase 6 (CMIP6), the first major science application of the model. Fig. 4 shows the global mean surface temperature for several of the key CMIP6 experiments. Table 2 lists the variants of CanESM5 which are being submitted to CMIP6. These include the "p1" and "p2" perturbed physics members of CanESM5 (see Section 2.5), and a version of the model with a different ocean biogeochemistry model, CanESM5-CanOE.

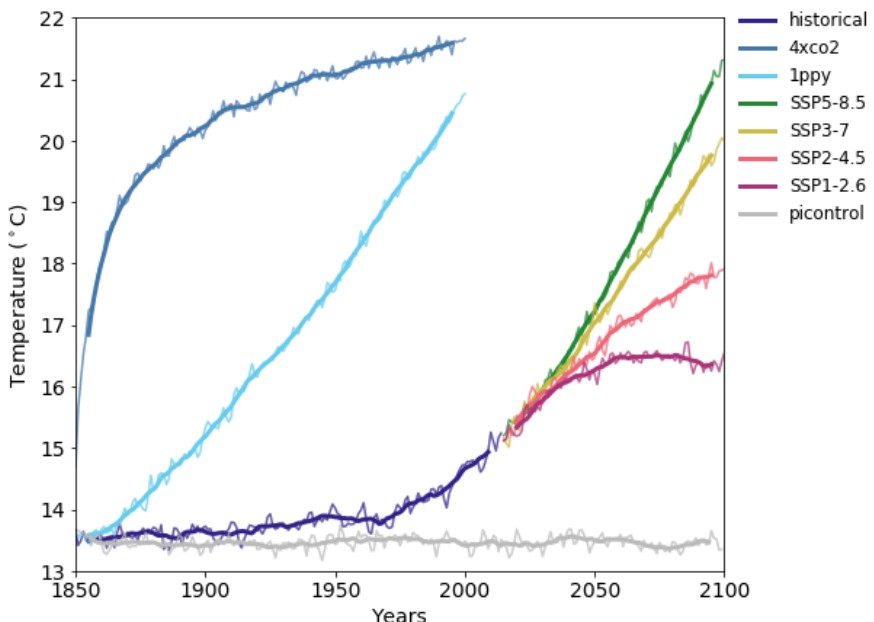

**Figure 4: Global average screen temperature in CanESM5 for the CMIP6 DECK experiments, as well as the historical and tier 1 SSP experiments (SSP5-85, SSP3-70, SSP2-45 and SSP1-26). Thick lines are the 11 year running means, thin lines are annual means.**

Table D1 lists the 20 CMIP6 endorsed MIPs in which CanESM5 is participating, and which model variants are being run for each MIP. The volume of simulation continues to grow, and will likely exceed 60,000 years. This is significantly more than the ~40,000 years of CMIP6 simulation estimated by Eyring et al. (2016). The major reason for this is that significantly larger ensembles have been produced than formally requested. For example, CanESM5 will submit at least 25 realizations for the historical and tier 1 SSP experiments, for each the "p1" and "p2" model variant, for a total of 50 realizations, significantly more than the single requested realization. The scientific value of such large initial condition ensembles has become evident (e.g. Kay et al., 2015; Kirchmeier-Young et al., 2017; Swart et al., 2018) and motivates this approach.





Individual historical realizations (ensemble members) of CanESM5 were generated by launching historical runs at 50 year
intervals off the piControl simulation. This is the same as the approach used to generate the five realizations of CanESM2,
which were submitted to CMIP5. The fifty year separation was chosen to allow for differences in multi-decadal ocean
variability between realizations. Below we discuss the properties of the model, including illustrations of the internal variability
generated spread across the historical ensemble. All results below are based on the CanESM5 p1 model variant.

**Table 2: Model variants**

| Model variant | Description |
|---|---|
| CanESM5 "p1" | CanESM5 realizations labelled as perturbed physics member 1 ("p1" in the variant label) have conservative remapping of wind-stress fields. The ocean biogeochemistry model is CMOC. |
| CanESM5 "p2" | CanESM5 realizations labelled as perturbed physics member 2 ("p2" in the variant label), use bilinear remapping of the wind-stress fields. A minor land-fraction change also occurs over Antarctica. The ocean biogeochemistry model is CMOC. |
| CanESM5-CanOE "p2" | CanESM5-CanOE is exactly the same physical model as CanESM5, but it uses the CanOE ocean biogeochemical model. All CanESM5-CanOE realizations use bilinear remapping of the wind-stress, and hence are labelled as perturbed physics member 2 ("p2" in the variant label). No "p1" variant is submitted. For physical climate purposes CanESM5 and CanESM5-CanOE may be treated as different realizations of the same model. |


**4 Stability of the pre-industrial control climate**
The characteristics and stability of the CanESM5 pre-industrial control climate are evaluated using 1000 years of simulation
from the CMIP6 piControl experiment, conducted under constant specified greenhouse gas concentrations and forcings for the
year 1850 (Eyring et al., 2016). Ideally, a climate model and all its subcomponents would exhibit perfect conservation of tracer
mass (e.g. water, carbon), energy and momentum, and would be run for long enough to achieve equilibrium. In this case we
would expect to see, on long term average, zero net fluxes of heat, freshwater and carbon at the interface between the
atmosphere, ocean and land surface, zero top of atmosphere net radiation, and constant long-term average temperatures or
tracer mass within each component. In reality however models are not perfectly conservative due to the limitations of numerical
representation (i.e. machine precision) as well as possible design flaws or bugs in the code, and models are generally not run
to perfect equilibrium due to computational constraints. Despite imperfect conservation or spin up, models can still usefully
be applied, as long as the drifts in the control run are small relative to the signal of interest, in our case historical anthropogenic
climate. Below we consider conservation and drift of heat, water and carbon in CanESM5 (Fig. 5).





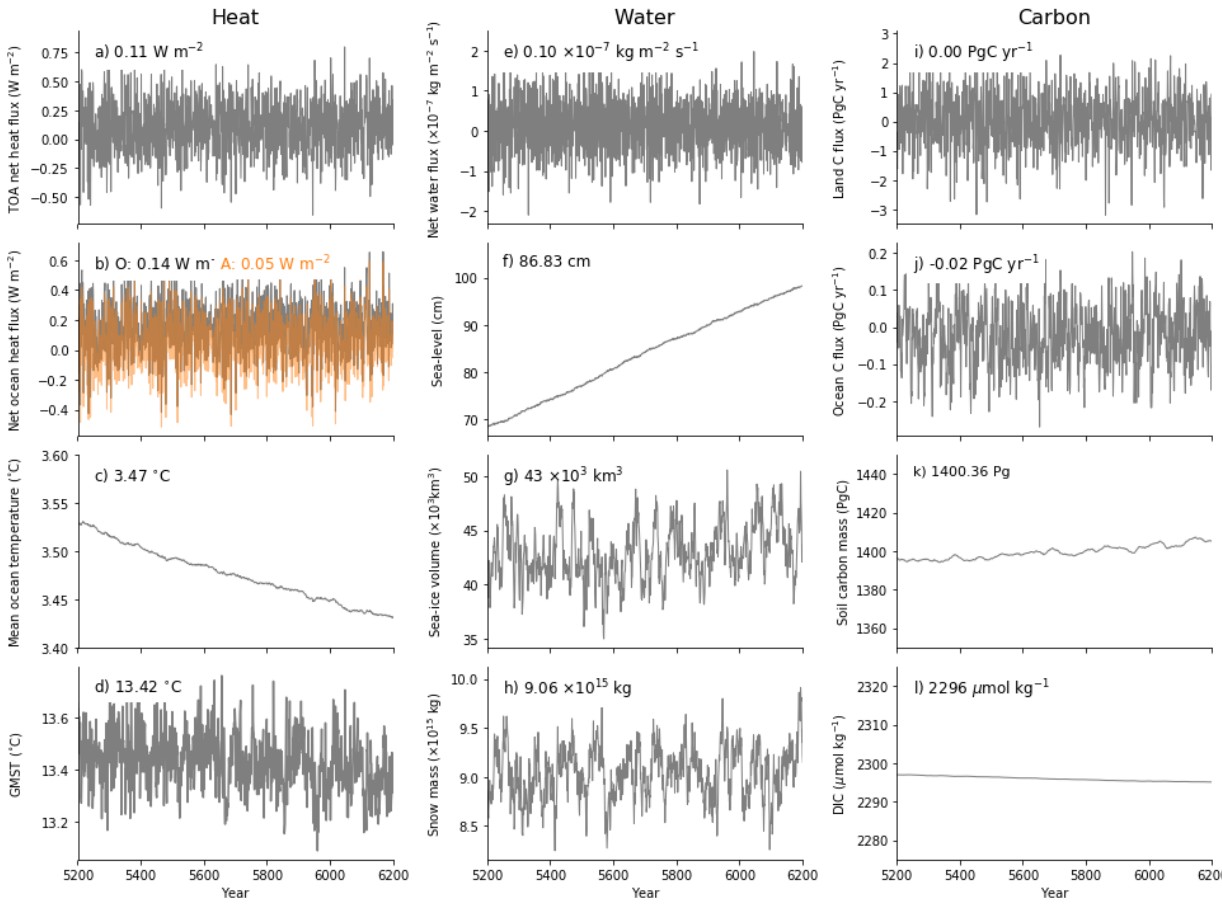


**Figure 5: Stability of the CanESM5 piControl run, showing (a) top of atmosphere net heat flux, (b) net heat flux at the surface of ocean; (c) volume averaged ocean temperature; (d) global mean screen temperature; (e) net freshwater input at the liquid ocean surface; (f) dynamic sea level; (g) global sea-ice volume; (h) global snow mass; (i) land-atmosphere carbon flux; j) ocean-atmosphere carbon flux. Heat fluxes in (a) and (b) are reported per metre squared of global area. The orange line in (b) is the heat flux computed at the bottom of the atmosphere, while the grey line is the heat flux computed at the surface of the liquid ocean (below sea-ice).**

450

The CanESM5 pre-industrial control shows a stable Top of Atmosphere (TOA) net heat flux of 0.1 Wm$^{-2}$ (fluxes positive down in m$^2$ of global area, Fig. 5a). The model is close to radiative equilibrium and this control net TOA heat flux is over an order of magnitude smaller than the signal expected from historical anthropogenic forcing (>1 Wm$^{-2}$). The global mean screen temperature is stable at around 13.4°C (Fig. 5d), indicating thermal equilibrium, and approximately in line with estimates of





the temperature in 1850. Half of the net TOA flux is passed from the atmosphere to the ocean (0.05 Wm$^{-2}$, Fig. 5b). With the conservative remapping in the coupler, the fluxes exchanged between components are identical to machine precision. However, the net heat flux received at the surface of the liquid ocean is 0.14 Wm$^{-2}$, almost three times higher than the heat flux passed from CanAM to NEMO (Fig. 5b). This discrepancy reflects a non-conservation of heat within the LIM2 ice model. Tests with an ice-free ocean do not suffer this problem. Nonetheless, the discrepancy is relatively small, and ice volume is stable. A further non-conservation occurs within the NEMO liquid ocean. Although the ocean receives a net heat flux of 0.14 Wm$^{-2}$, the volume averaged ocean is cooling at a rate equivalent to a flux of 0.05 Wm$^{-2}$ (Fig. 5c) implying a total non-conservation of heat in the liquid ocean of about 0.2 Wm$^{-2}$. Conservation errors of this order are well known in NEMO v3.4.1, likely arise from the use of the linear free surface (Madec et al., 2012), and have been seen in previous coupled models using NEMO (Hewitt et al., 2011). Despite this, the volume averaged ocean temperature drift in CanESM5 is about half the size of the drift in CanESM2. Furthermore the lack of ocean heat conservation in CanESM5 is roughly constant in time, and appears to be independent of the climate (not shown).

At the liquid ocean surface, a small net freshwater flux results in a freshening trend, and a sea-level rise of about 24 cm over 1,000 years (Fig. 5e, f). This rate of drift is more than 20 times smaller than the signal of anthropogenic sea-level rise. The LIM2 ice model appears to be the source of non-conservation: the net freshwater flux provided from CanAM is very close to zero, about six times smaller than that noted above (24 cm / 1000 years). Snow and ice volume are stable, not exhibiting any long term drift, yet they are subject to considerable decadal and centennial scale variability (Fig. 5g, h).

Atmosphere-land carbon fluxes average to zero, and carbon pools within CTEM are stable (Fig. 5i, k). The net ocean carbon flux is fairly close to zero, but remains slightly negative on average at -0.02 Pg yr$^{-1}$ despite a multi-millennial spin up (Fig. 5j). The total mass of dissolved inorganic carbon in the ocean decreases very slightly as a result (Fig. 5l). The rate of ocean carbon drift is approximately an order of magnitude smaller than the modern day anthropogenic signal of ocean carbon uptake (>2 Pg yr$^{-1}$). The drifts identified above are all far smaller than would be expected from anthropogenically forced trends, confirming that the model is suitably stable to evaluate centennial scale climate change. In the following section, we consider the ability of the model to reproduce large scale features of the observed historical climate.

## 5 Evaluation of historical mean climate

In this section we use the CMIP6 historical simulations (Eyring et al., 2016) of CanESM5 "p1", focusing on climatologies computed over 1981 to 2010, unless otherwise noted.

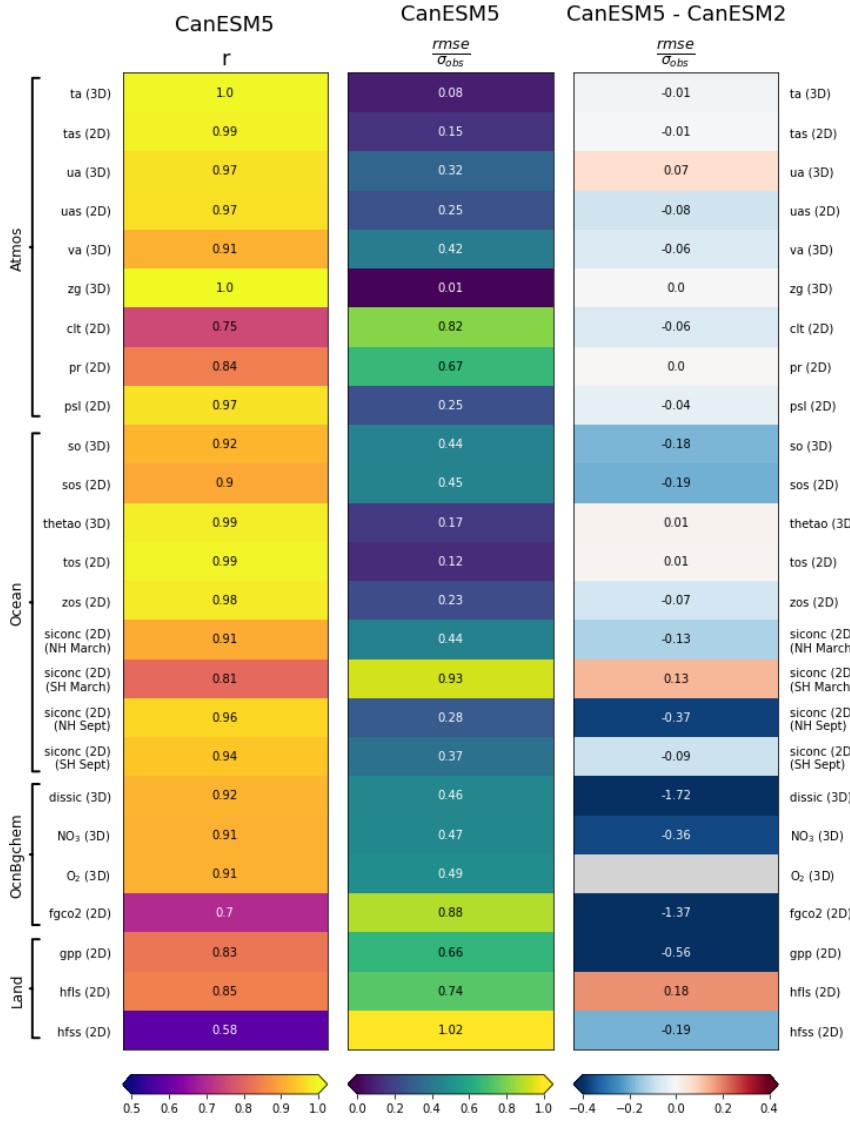

484

**Figure 6: Summary statistics quantifying the ability of CanESM to reproduce large scale climate features. Shown are**

**the correlation coefficient (r) between the simulated and observed spatial patterns, the Root Mean Square Error**

**(RMSE) normalized by the (observed spatial) standard deviation (σ), and the difference in normalized RMSE**

**between CanESM5 and CanESM2. The spatial quantities represent temporal means over 1981 to 2010, except as**

**noted in appendix F. Variables are labelled according to the names in the CMIP6 data request.**



## 5.1 Overall skill measures

The ability of CanESM5 to reproduce observed large scale spatial patterns in the climate system is quantified using global summary statistics computed over the 1981 to 2010 mean climate (Fig. 6). Shown are the correlation coefficient between CanESM5 and observations (r), the Root Mean Square Error (RMSE) normalized by the observed (spatial) standard deviation (σ), and the change in normalized RMSE between CanESM2 and CanESM5. The statistics are weighted by grid cell area for 2D fields, volume for 3D ocean fields, and by area and pressure for 3D atmospheric variables. In general CanESM5 successfully reproduces many observed spatial patterns of the surface climate, interior ocean, and the atmosphere, with correlation coefficients between the model and observations generally above 0.8. Some exceptions are the total cloud fraction (clt, r=0.75), atmosphere-ocean $CO_2$ flux (fgco2, r=0.7) and the surface sensible heat flux (hfss, r=0.58).

For most variables, normalized RMSE has decreased in CanESM5 relative to CanESM2, indicating an improvement in the ability of the new model to reproduce observed climate patterns over its predecessor. The largest improvements were seen for ocean biogeochemistry variables, while small increases in error were seen for 3D distribution of zonal winds (ua), sea surface temperatures (tos), the March distribution of sea-ice in the Southern Hemisphere (siconc), and surface latent heat flux (hfls). In the following sections individual realms are examined, with a closer look at regional details and biases.

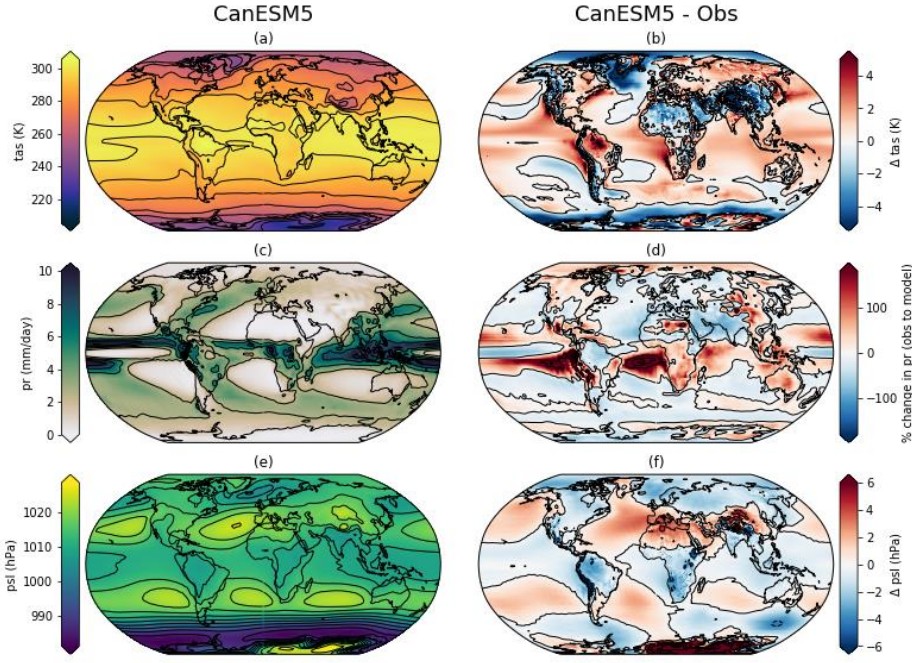

**Figure 7: Climatologies over 1981 to 2010 of (a) surface air temperature, (c) precipitation and (e) sea-level pressure in CanESM5, and their bias from (b) ERA5, (d) GPCP and (f) ERA5 over the same period.**

**5.2 Atmosphere**

CanESM5 reproduces the large scale climatological features of surface air temperatures, precipitation and sea-level pressure, though significant regional biases exist (Fig. 7). CanESM5 is significantly colder than observed over sea-ice covered regions (Fig. 7a, b), noticeable in the Southern Ocean, and most obviously in the region surrounding the Labrador sea, which has extensive seasonal sea-ice cover in CanESM5 (see below). The Tibetan plateau, the Sahara and the broader North Atlantic Ocean are also cooler than observed. Warm biases exist over the eastern boundary current systems (Benguela, Humboldt, and California); over the Amazon, eastern North America, much of Siberia, and over broad regions of the tropical and subtropical oceans.

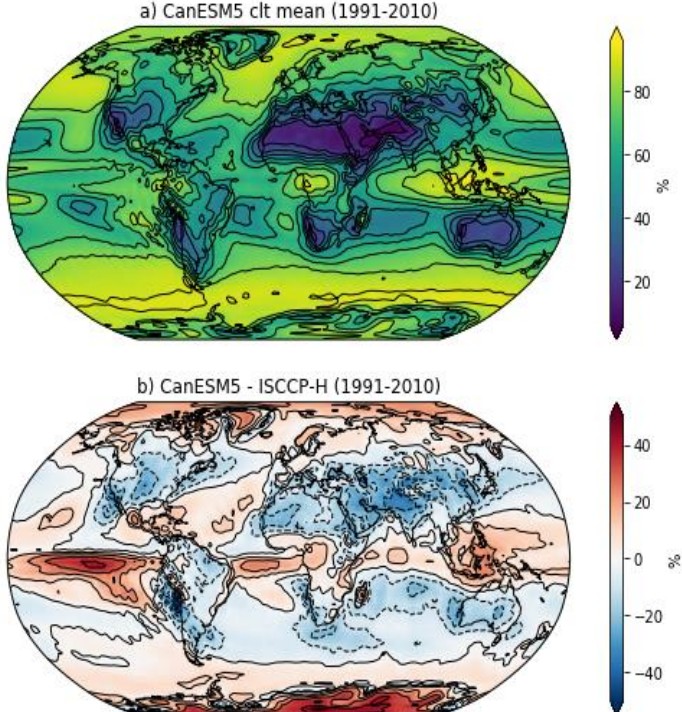

**Figure 8: Cloud fraction in (a) CanESM5 and (b) the bias with respect to ISCPP-H satellite based observations.**

Precipitation biases vary in sign by region (Fig. 7d). The largest relative (to mean) biases are excessive simulated precipitation over the eastern Pacific and Atlantic oceans, between the equator and extending into the southern subtropics. The largest land biases are excessive precipitation over much of sub-Saharan Africa, Southeast Asia, Canada, and Peru-Chile. In contrast western Asia, Europe, the North Atlantic and the subtropical to high-latitude Southern Oceans have too little simulated precipitation. The large scale pattern of sea-level pressure is captured by CanESM5 (Fig. 7e). Biases relative to ERA5 are





largest over the high elevations of Antarctica (Fig. 7f), possibly reflecting differences in the extrapolation of surface pressure
to sea-level.
Relative to ISCCP-H (Young et al, 2018), version 1.00 (Rossow et al, 2016) the total cloud fraction in CanESM5 is
overestimated along the equator, particularly in the eastern tropical Pacific (Fig 8). Too large cloud fraction is also found over
Antarctica and the Arctic. Underestimates of total cloud fraction occur over most other land areas, with the largest
underestimates over Asia and the Himalayas.
Zonal mean sections of air temperature for the DJF and JJA seasonal means are shown in Fig. 9. In both seasons, CanESM5
is biased warm relative to ERA5 near the tropopause, across the tropics and subtropics. Warm biases also occur in the
stratosphere, notably near 60°S above 50 hPa in JJA. Cold biases exist from the subtropics to the high latitudes, where they
reach from the surface to the stratosphere, and are strongest in the winter season.

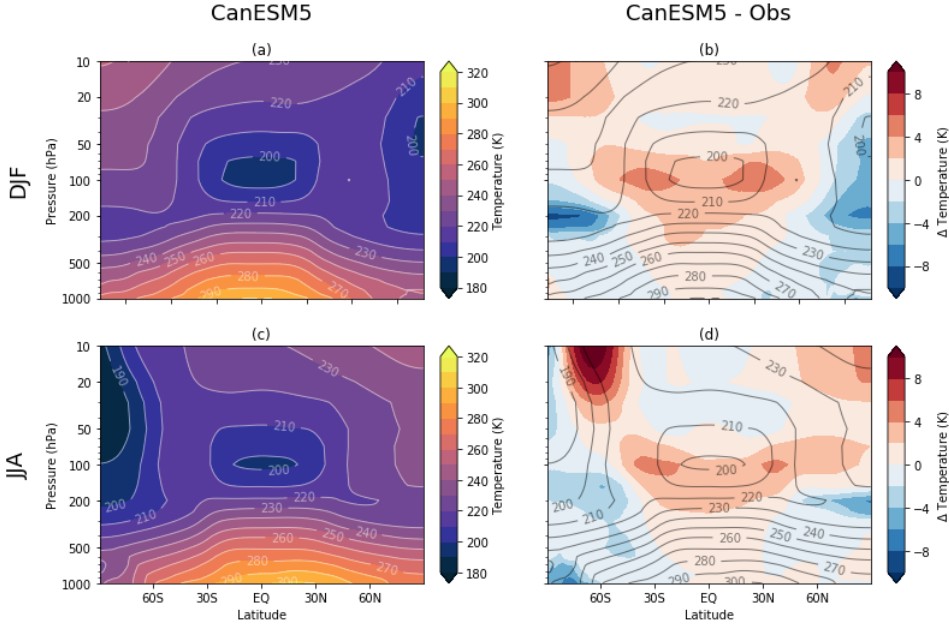

**Figure 9: Zonal mean temperature in CanESM5 (a, c) and bias relative to ERA5 (b, d) over 1981-2010, for the DJF (a, b) and JJA (c, d) seasons.**



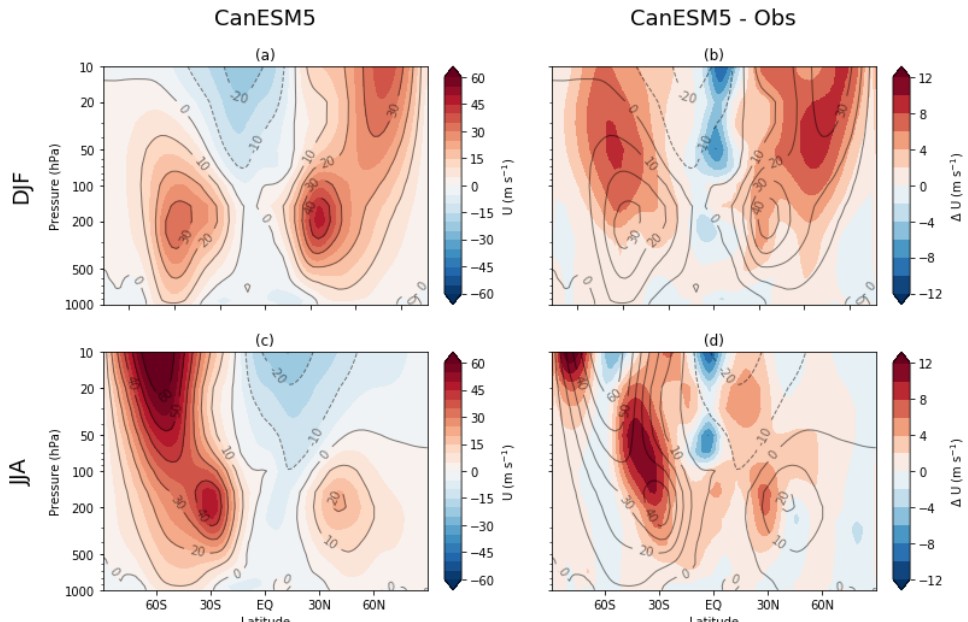

**Figure 10: Zonal mean zonal winds (a, c) and bias relative to ERA5 (b, d) over 1981-2010, for the DJF (a, b) and JJA (c, d) seasons.**

Zonal mean zonal winds are compared to ERA5 in Fig. 10 for DJF and JJA. The westerly jets in CanESM5 are biased strong, particularly aloft and in the winter hemisphere. Surface zonal winds in CanESM5 are only slightly stronger than observed, and are significantly improved over those in CanESM2 (Fig. 11), which were too strong, particularly over the Southern Hemisphere westerly jet.

**5.3 Land physics and biogeochemistry**

Figures 12 and 13 compare the geographical distribution and zonal averages of gross primary productivity (GPP), and latent and sensible heat fluxes over land with observation-based estimates from Jung et al. (2009). The zonal averages of GPP, and latent and sensible heat fluxes compare reasonably well with observation-based estimates although the latent heat fluxes are somewhat higher especially in the southern hemisphere as discussed below (Fig. 13). Figure 12 shows the biases in the simulated geographical distribution of these quantities. In the tropics biases in GPP, and latent and sensible heat fluxes, broadly correspond to biases in simulated precipitation compared to observation-based estimates (shown in Fig. 7).

563

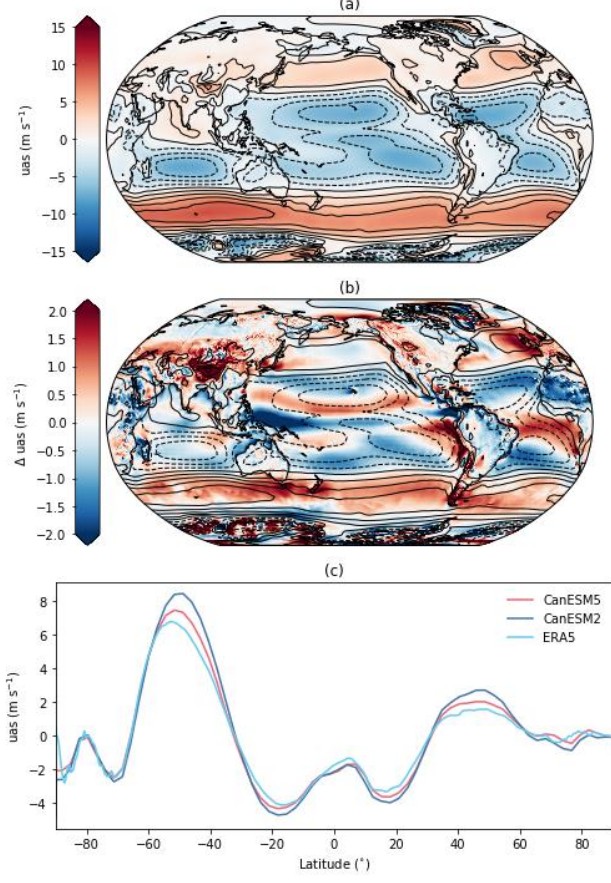

**Figure 11: Zonal surface winds in (a) CanESM5, (b) the bias relative to ERA5 and (c) zonal-mean zonal surface winds in CanESM2, CanESM5 and ERA5.**

Generally over tropics, as would be intuitively expected, the sign of GPP and latent heat flux anomalies are the same since they are both affected by precipitation in the same way. Sensible heat flux is expected to behave in the opposite direction compared to GPP and latent heat flux in response to precipitation biases. For example, simulated GPP and latent heat fluxes are lower, and sensible heat fluxes higher in the north eastern Amazonian region because simulated precipitation is biased low (Fig. 7). The opposite is true for almost the entire African region south of the Sahara desert and most of Australia. Here simulated precipitation that is biased high, compared to observations, results in simulated GPP and latent heat flux that are higher and sensible heat flux that is lower than observation-based estimates. At higher latitudes, where GPP and latent heat flux are limited by temperature and available energy, the biases in precipitation do not translate directly into biases in GPP and latent heat flux as they do in the tropics.

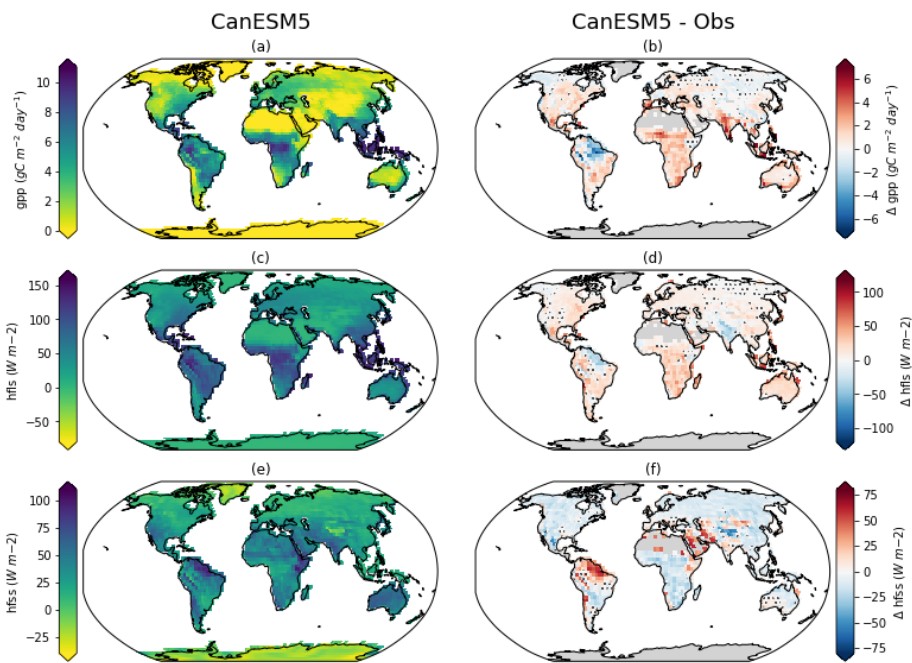

**Figure 12: Time-mean values of (a) gross primary productivity (GPP), (c) latent heat flux (HFLS), and (e) sensible heat flux (HFSS) from CanESM5 (r1i1p1f1) (left-hand column) and the corresponding biases with respect to observation-based reference data presented in Jung et al. (2009) (GBAF) (right- hand column). Black dots mark grid cells where biases are not statistically significant at the 5% level using the two-sample Wilcoxon test.**

The biases in simulated climate imply that simulated land surface quantities will also be biased which make it difficult to assess if the underlying model behaviour is realistic. This limitation can be alleviated to some extent by looking at the functional relationships between a quantity and its primary climate drivers. This technique works best when a land component is driven offline with meteorological data. In a coupled model, as is the case here, land-atmosphere feedbacks can potentially worsen a model's performance by exaggerating an initial bias. For example, low model precipitation can be further reduced due to feedbacks from reduced evapotranspiration some of which is recycled back into precipitation. Figure 14 shows the functional relationships between GPP and temperature, and GPP and precipitation, for both model and observation-based estimates. The observations-based temperature and precipitation data used in these plots are from CRU-JRA reanalysis data that were drive participating terrestrial ecosystem models in the TRENDY Intercomparison for the 2018 Global Carbon Budget (Le Quéré et al., 2018). Figure 14 shows that GPP increases both with increases in precipitation (as would be normally expected) and temperature except at mean annual values above 25 °C when soil moisture limits any further increases. This threshold emerges both in the model and the observation-based functional relationships. With the caveat mentioned above, the functional relationships of GPP with temperature and precipitation based on simulated data compare reasonably well with those based on

observation-based data, although the simulated GPP relationship with precipitation compares much better to its observation-
based relationship than that for temperature.

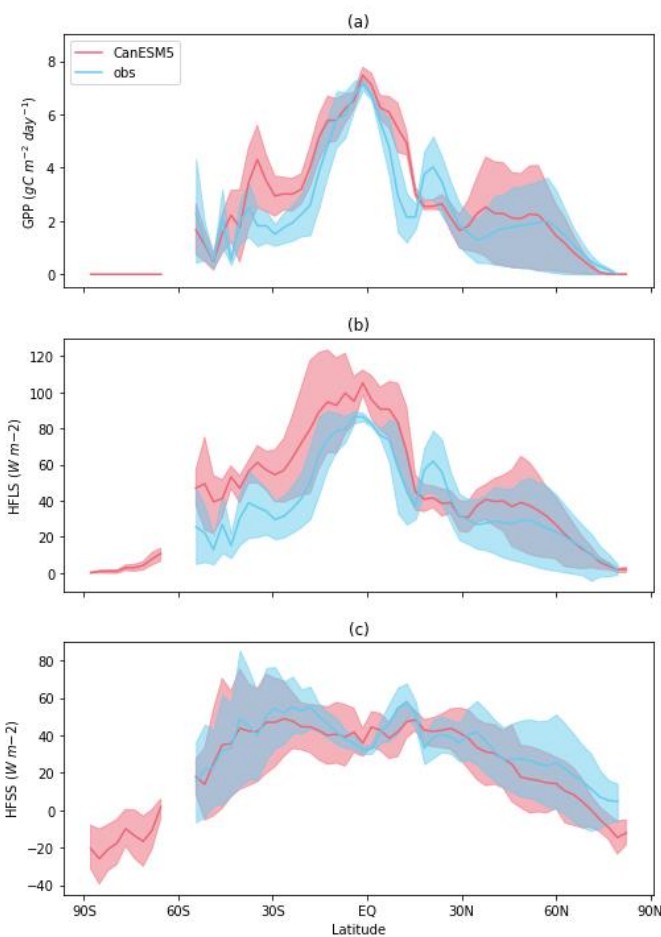


**Figure 13: Zonal mean values of (a) GPP, (b) HFLS, and (c) HFSS for CanESM5 (r1i1p1f1) (black) and reference**
**data (red) from Jung et al. (2009). The shading presents the corresponding inter-quartile range (IQR).**

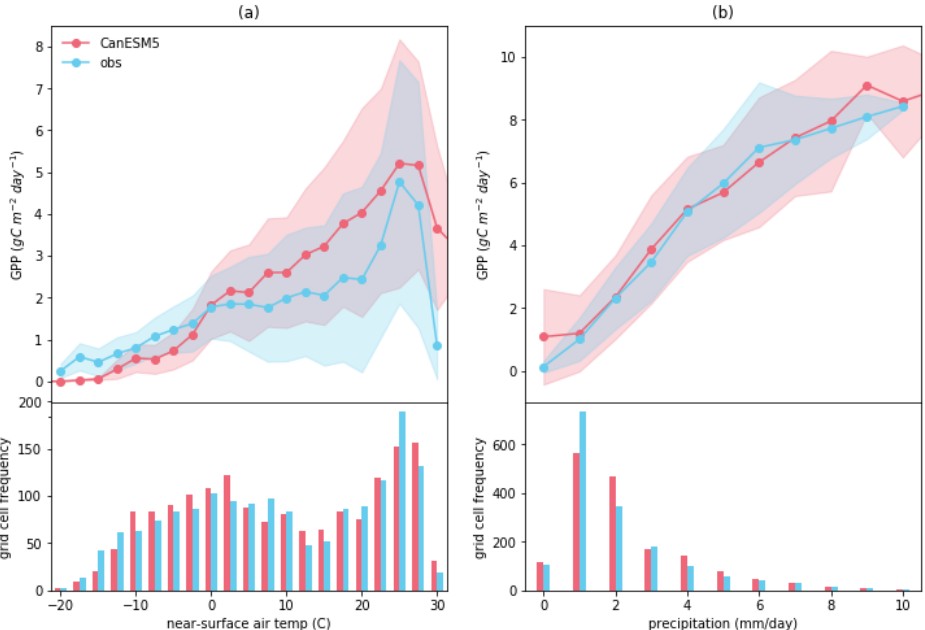

**Figure 14: Functional response of GPP to (a) near-surface air temperature and (b) surface precipitation for CanESM5 (r1i1p1f1) (black) and reference data (red) from Jung et al. (2009) (GBAF). The shading presents the corresponding standard deviation. The grid cell frequencies are shown in the lower part of each plot.**

**5.4 Physical ocean**

CanESM5 reproduces the observed large scale features of sea surface temperature (SST), salinity (SSS) and height (SSH) (Fig. 15). The largest SST biases are the cold anomalies south east of Greenland and in the Labrador Sea (Fig. 15b). These negative SST biases are associated with excessive sea-ice cover, described further below, and with the surface air temperature biases mentioned above. Positive SST biases are largest in the Eastern Boundary Current upwelling systems, as for surface air temperatures.

Sea surface salinity biases are largest, and positive, around the Arctic coastline, potentially indicating insufficient runoff in this region (Fig. 15d). Negative annual mean SSS biases occur under the region associated with excessive March sea-ice in the Labrador Sea, and are also found in seas of the maritime continent. Sea-surface height (SSH) is shown as an anomaly from the (arbitrary) global mean (Fig. 15e). Significant SSH biases are associated with the positions of western boundary currents, noticeably for the Gulf Stream and Kuroshio current (Fig. 15f). CanESM5 has too low SSH around Antarctica, and too high SSH in the southern subtropics, with an excessive SSH gradient across the Southern Ocean. This SSH gradient is associated with the geostrophic flow of the Antarctic Circumpolar Current (ACC). The ACC in CanESM5 is vigorous with 190 Sv of transport through Drake Passage. This is larger than observational estimates which range up to 173.3 +/- 10.7 Sv (Donohue et



al, 2016). In CanESM5 the ACC also exhibits a pronounced, centennial scale variability of about 20 Sv, which is also evident
in the piControl simulation (not shown).


**Figure 15. Sea surface (a) temperature, (c) salinity and (e) height averaged over 1981 to 2010, and their biases relative to World Ocean Atlas 2009 (b, d), and the AVISO mean dynamic topography (f).**


The CanESM5 interior distributions of potential temperature and salinity are well correlated with observations (Fig. 6). In the
zonal mean, potential temperature biases are largest within the thermocline, which is warmer than observed, particularly near
50°N (Fig. 16a, b). The deep ocean, the Southern Ocean south of 50°S and the Arctic Ocean are cooler than observed. The
pattern of excessive heat accumulation in the thermocline is very similar to the pattern of bias seen in CMIP5 models on
average (Flato et al., 2013 their Fig. 9.13). The major salinity bias is of excessive fresh waters in the Arctic near 250 m, also
typical of the CMIP5 models (Fig. 16d). Sea-surface salinities showed the Arctic to be too salty, but this bias is confined to
near the surface, and at all depths below the immediate surface layer the Arctic Ocean is too fresh. The zonal mean salinity
also shows a positive salinity bias near 40°N, associated with the Mediterranean outflow.

The Meridional Overturning Circulation in the global ocean, and the Indo-Pacific, as well and Arctic-Atlantic basins is shown
in Fig. 17. The global overturning streamfunction shows the expected major features: an upper cell with clockwise rotation,
connecting North Atlantic Deepwater formation to low latitude and Southern Ocean upwelling; a vigorous Deacon cell in the
Southern Ocean (as a result of plotting in z-coordinates); a lower counter clockwise cell of Antarctic Bottom Water, and



vigorous near-surface cells in the subtropics. The upper cell overturning rate at 26°N in the Atlantic is estimated to be 17±4.4
Sv from the RAPID observational array (McCarthy et al. 2015). CanESM5 produces an Atlantic overturning rate of 12.8 Sv
at 26°N, below the mean but within the range measured by RAPID.

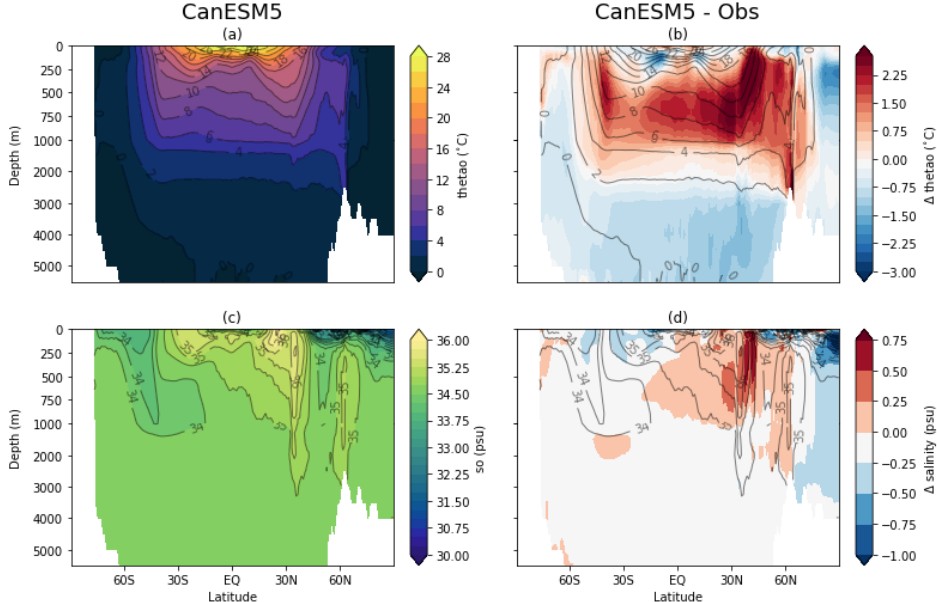


**Figure 16: CanESM5 zonal mean ocean (a) potential temperature, (c) salinity averaged over 1981 to 2010, and their**
**biases from World Ocean Atlas 2009 (b, d). Note the depth-scale on the y-axis is non-uniform.**



Closely connected to the MOC is the rate of northward heat transport by the ocean (Fig. 18). CanESM5 produces the expected
latitudinal distribution of heat transport, but consistent with a weak MOC, slightly underestimates the transport at 24°N, relative
to the inverse estimate of Ganachaud and Wunsch (2003). To the north and south, CanESM5 ocean heat transport falls within
the observational uncertainties. The MOC and heat transport in CanESM5 are similar to those in CanESM2, as reported in
Yang and Saenko (2012)







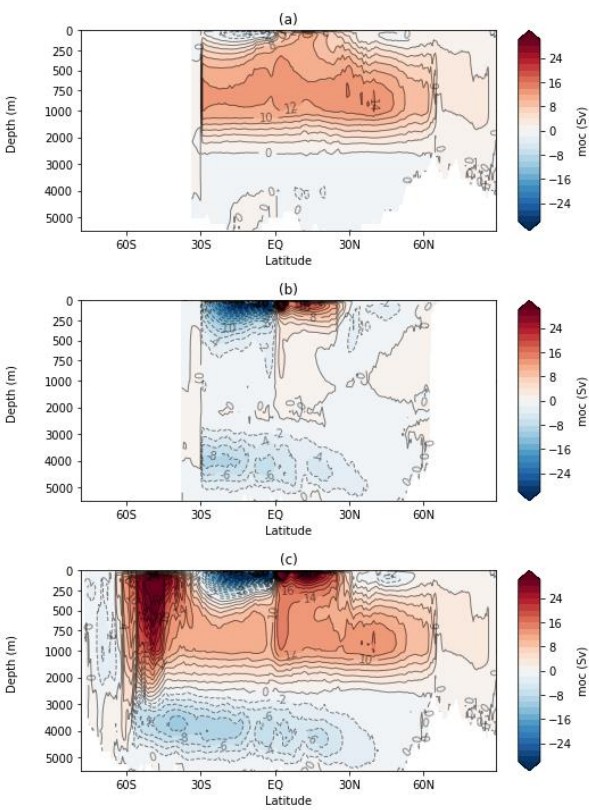


**Figure 17: CanESM5 residual meridional overturning circulation in the Atlantic (a), Indo-Pacific (b) and global (c)**

**oceans, averaged over 1981 to 2010 including all resolved and parameterized advective processes. Note the depth-**

**scale on the y-axis is non-uniform.**


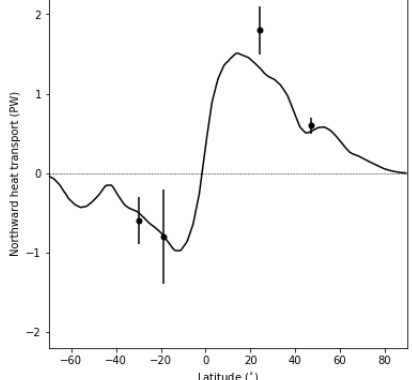


**Figure 18: Northward heat transport in the global ocean in CanESM5 (in Petawatts), with error bars showing**

**the inverse estimate of Ganachaud and Wunsch (2003).**


### 5.5 Sea-ice

The seasonal cycle of sea-ice extent and volume are shown in Fig. 19. A major change from CanESM2 is seen in the sea-ice
volume (Fig. 19b, d). CanESM2 simulated very thin ice, and had about 40% less Northern Hemisphere (NH) ice volume than
in the PIOMAS reanalysis. By contrast, CanESM5 has a larger NH ice volume than in CanESM2 and in PIOMAS (Fig. 19b).
The amplitude and phase of the annual cycle in NH sea-ice volume in CanESM5 is similar to PIOMAS (Fig. 19b). In the
Southern Hemisphere, CanESM5 also has a larger sea-ice volume and seasonal cycle far more consistent with the GIOMAS
reanalysis product than CanESM2 (Fig. 19d).

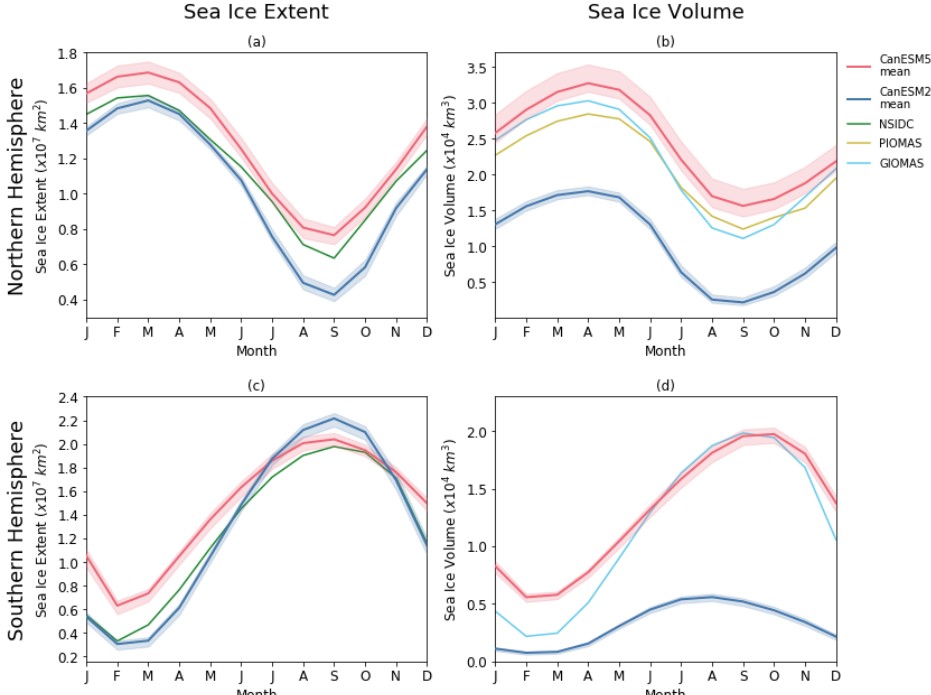

**Figure 19: Seasonal cycles of sea-ice extent (a, c) and volume (b, d) in the Northern (a, b) and Southern (c, d) hemispheres averaged over 1981 to 2010. Results are shown for CanESM2, CanESM5, the NSIDC satellite based observations, and the PIOMAS and GIOMAS reanalyses.**

While CanESM2 significantly underestimated NH sea-ice extent relative to satellite based observations, CanESM5 generally
overestimates the extent (Fig. 19a). The NH sea-ice extent biases are largest in the winter and spring. During the March
maximum, excessive sea-ice is present in the Labrador Sea and east of Greenland (Fig. 20a). In the summer and fall, the net



NH extent bias is far smaller (Fig. 20c), and results from a cancellation between lower than observed concentrations over the
Arctic basin and larger than observed concentrations around northeastern Greenland. Southern Hemisphere sea-ice extent
biases are largest during the early months of the year, and in March the positive concentration biases are focused in the
northeastern Weddell and Ross Seas (Fig. 20b). In September SH concentration biases between CanESM5 and the satellite
observations are focused around the northern ice-edge, and are of varying sign (Fig. 20d).

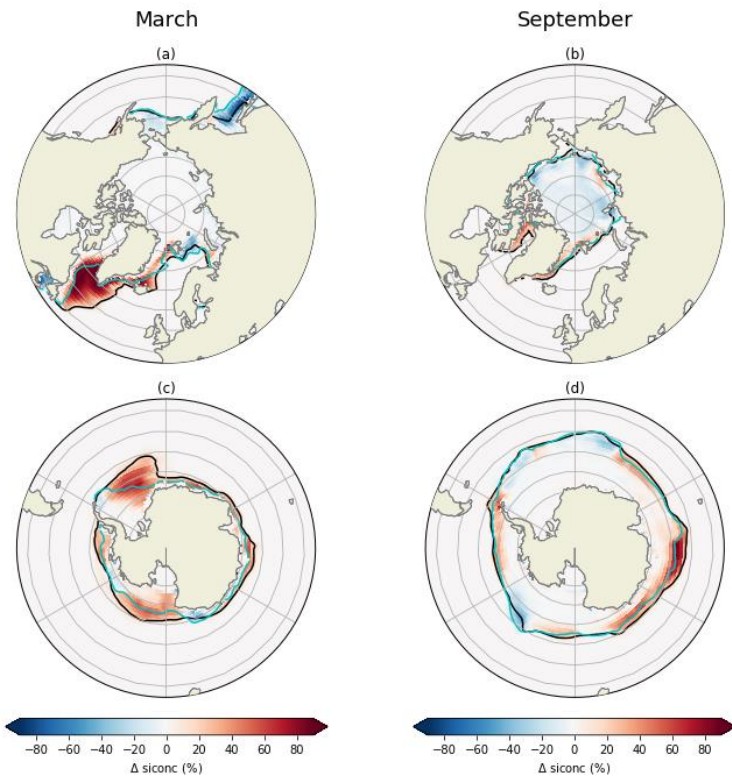

**Figure 20: Sea-ice concentration biases between CanESM5 and NSIDC climatologies for the months of March (a, c)**
**and September (b, d), in the Northern (a, b) and Southern (c, d) hemispheres. The solid black contour marks the ice-**
**edge (15% threshold) in CanESM5, and the teal line marks the ice-edge in the observations. Biases are based on the**
**1981 to 2010 climatology.**

**5.6 Ocean biogeochemistry**
The standard configuration of CanESM5 has a significantly improved representation of the distribution of ocean
biogeochemical tracers relative to CanESM2, despite using the same biogeochemical model (CMOC). For the three-
dimensional distributions of Dissolved Inorganic Carbon (DIC) and NO3, and the surface $CO_2$ flux, the Root Mean Square



Error (RMSE), relative to observed distributions was reduced by over a factor of two (Fig.6). Ocean only simulations, whereby
NEMO was driven by CanESM2 surface forcing via bulk formulae, show similar skill to the CanESM5 coupled model. From
this we infer that changes in interior ocean circulation, rather than boundary forcing, are responsible for the improved
representation of biogeochemical tracer distributions.

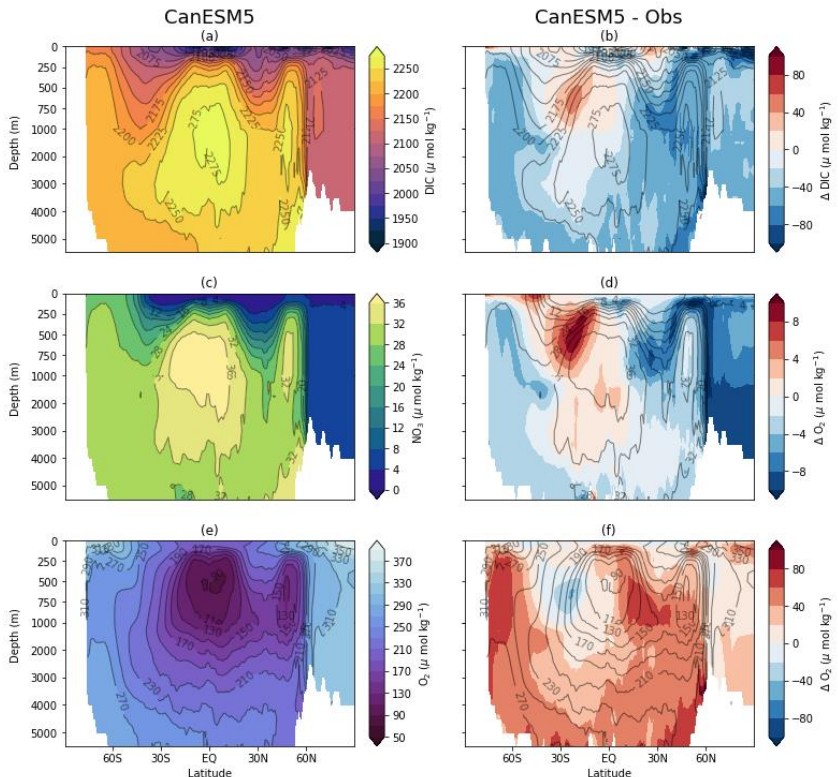


**Figure 21: Zonal mean sections of (a) Dissolved Inorganic Carbon, (c) NO₃, and (e) O₂ in CanESM5 averaged over**
**1981 to 2010, and their biases relative to GLODAP v2 (b, d, f). Note the depth-scale on the y-axis is non-uniform.**

In CanESM5 the zonal mean DIC concentration simulated by CMOC is generally lower than observed, by amounts reaching
up to about 5% (Fig. 21a, b). One exception to this is in the SH subtropical thermocline, on the northern flank of the Southern
Ocean, which shows positive DIC biases between 250 and 1000 m. This area is also one of positive nitrate biases, whose
magnitude is close to 30% (Fig. 21d). Elsewhere zonal-mean $NO_3$ concentrations are generally too low, particularly in the NH
thermocline and the Arctic. CanESM5 has higher than observed concentrations of zonal mean $O_2$ (Fig. 21f). As expected from
saturation, biases are largest in the Southern and abyssal ocean, where CanESM5 is colder than observed. However, positive
$O_2$ biases also occur at the base of the thermocline in the NH, where CanESM5 is too warm, suggestive of a biological origin.

The zonal mean $NO_3$ biases identified at the thermocline level above are the result of partially cancelling biases between the
Pacific and Atlantic basins (not shown). The Atlantic has negative $NO_3$ biases, largest near 1000 m. Meanwhile, there is an
excessive accumulation of $NO_3$ centered at the base of the eastern Pacific thermocline. This buildup occurs due to the simplified
parameterization of denitrification in CMOC. Within each vertical column, the amount of denitrification is set to balance the
rate of nitrogen fixation, and is distributed vertically proportional to the detrital remineralization rate. In reality nitrogen
fixation and denitrification are not constrained to balance within the water column at any one location, but rather denitrification
proceeds within anoxic areas. A prognostic implementation of denitrification implemented into CanOE resolves this bias, and
will be discussed further in an upcoming article within this special issue.

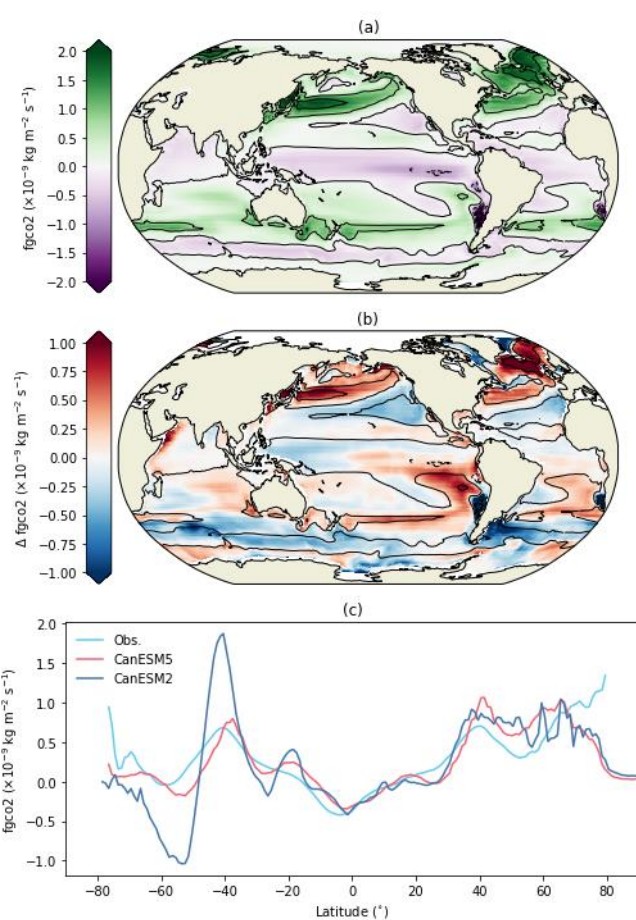


**Figure 22: Ocean atmosphere flux of $CO_2$ in (a) CanESM5 averaged over 1981 to 2010 (b) from Landschutzer (2009),**
**and (c) zonal mean $CO_2$ flux in CanESM2, CanESM5 and Landschutzer (2009) data.**




The atmosphere-ocean $CO_2$ flux pattern in CanESM5 correlates significantly better with estimates of the observed flux than CanESM2 (Fig. 6). The largest departures from the observations are positive biases in the southeastern Pacific, northwest Pacific and northwest Atlantic (Fig. 22b). These are compensated by negative biases in the Southern Ocean and mid-latitude northeast Pacific. In the zonal mean, CanESM2 had a large flux dipole in the Southern Ocean, which is significantly reduced in CanESM5, and attributable to improved circulation in the new NEMO ocean model and a reduction in Southern Ocean wind speed biases in CanAM5 (Fig. 22c).

### 5.7 Modes of climate variability

### 5.7.1 El-Niño Southern Oscillation

The El-Niño Southern Oscillation (ENSO) is a key component of climate variability on seasonal and interannual timescales. To evaluate CanESM5's representation of ENSO, the NINO3.4 index (average monthly SST anomaly in the region bounded by 5S, 5N, 170W, 120W) from the first 10 historical ensemble members is compared against HadISST. The skill of CanESM5 at representing the local and remote effects of ENSO is evaluated by correlating SST anomalies with the resulting NINO3.4 index (Fig. 23a, b). Within the equatorial Pacific, a positive ENSO event in CanESM5 leads to an increase in SSTs across the entire basin whereas observations show negative SST anomalies in the western basin and positive anomalies in the central and eastern Pacific. ENSO in CanESM5 also has weaker teleconnections. The SST within the subtropical North and South Pacific gyres are more weakly anticorrelated to ENSO than observed. HadISST shows a negative North Atlantic Oscillation like pattern associated with ENSO, which is not present in CanESM5. The SST teleconnection in the tropical Indian and Atlantic Oceans is well represented by the model.

The spectral peak in the historical ensemble members (Fig. 23c) occurs at around 3-5 years in general agreement with observations. Variability on decadal time-scales has a large spread between ensemble members likely due to differences in the strength of warming trends over the historical period. Higher frequency variability at monthly to seasonal timescales is significantly lower than observed. The lower monthly variability can also be seen by examining month-by-month interannual variability of NINO3.4 (Fig. 23d). While January remains the month of peak variability, overall the annual cycle of NINO3.4 variability is weaker in CanESM5. In observations, ENSO variability is at its minimum between April and June but in CanESM5 the minimum variability (depending on the ensemble member) tends to be between July and September.

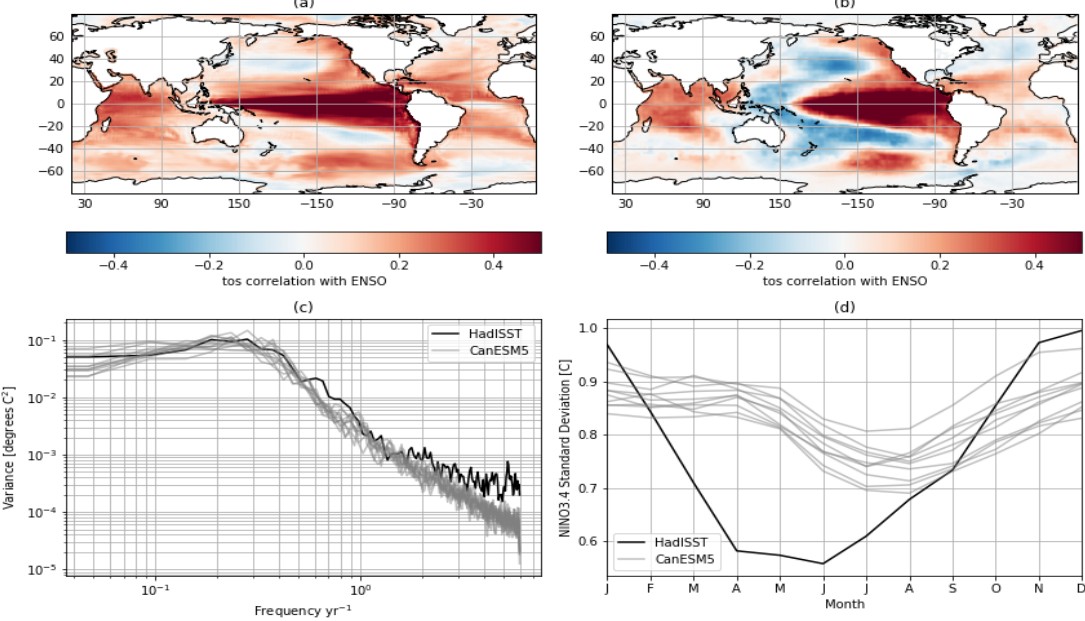

**Figure 23: Characteristics of the El Niño Southern Oscillation (ENSO) from and the HadISST observational product. Spatial maps in (a) and (b) are the regression of the SST monthly anomalies from 1850-2014 against the NINO3.4 index from (a) CanESM5 (historical ensemble member r1i1p1f1) and (b) from HadISST. Temporal variability is summarized as power spectra (c) of the NINO3.4 index from HadISST and ten historical ensemble members and the interannual variability of the NINO3.4 index by month (panel d) for CanESM5 and HadISST.**

### 5.7.2 Annular Modes

The Northern Annular Mode is computed as the first EOF of extended winter (DJFM) sea level pressure north of 20°N for CanESM5 and ERA5 (Fig. 24 a, b). The correlation between the CanESM5 and ERA5 patterns is 0.95. Despite the high degree of coherence, some differences between the model pattern and reanalysis are evident (Fig. 24). For example CanESM5 has a positive centre in the north Pacific, not seen in ERA5, and the positive pattern across the North Atlantic is less continuous in CanESM5. This is a typical model bias (e.g. Bentson et al., 2013). The first EOF in CanESM5 also explains about 8% more variance than in the reanalysis.

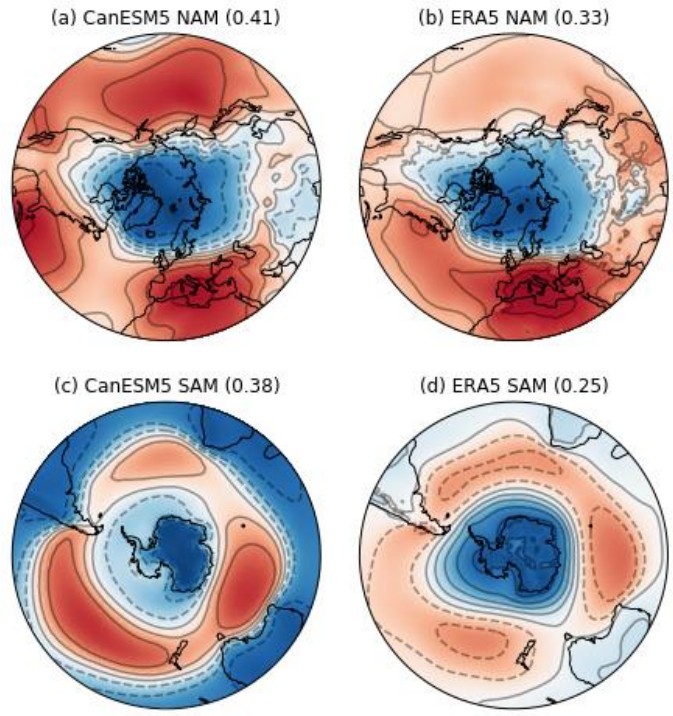

(a) CanESM5 NAM (0.41)     (b) ERA5 NAM (0.33)

(c) CanESM5 SAM (0.38)     (d) ERA5 SAM (0.25)

**Figure 24: First Empircal Orthogonal Functions (EOFs) of sea-level pressure north of 20N (a, b), and south of 20S (c, d), representing the Northern Annual Mode and Southern Annular Mode respectively. The NAM is based on the extended winter DJFM season, and the SAM is based on monthly sea-level pressure. Results are shown for CanESM5 (a, c) and ERA5 (b, d), and the amount of variance explain by each EOF is given in brackets. The color scale is arbitrary.**

The Southern Annular Mode is the dominant mode of climate variability in the Southern Hemisphere, with significant influences on atmospheric circulation, precipitation, and the Southern Ocean. We compute the SAM pattern as the first EOF of sea level pressure south of 20°S. The CanESM5 and ERA5 pattern correlation is 0.7. In CanESM5, the first EOF accounts for 13% more variance than in the reanalysis. Despite such biases, these results confirm that CanESM5 captures the principal modes of tropical and mid-latitude climate variability.

## 6 Climate response to forcing

### 6.1 Response to CO2 forcing

The global mean screen temperature change under the idealized CMIP6 DECK experiments "abrupt-4xCO2" and "1pctCO2" are shown in Fig. 4. From these simulations, three major benchmarks of the model's response to CO2 forcing can be quantified (Table 3).





**Table 3: Key sensitivity metrics: Transient Climate Response (TCR), Transient Climate Response to Cumulative Emissions (TCRE), and Equilibrium Climate Sensitivity (ECS).**

| Model | TCR (K) | TCRE (K/EgC) | ECS (K) |
|---|---|---|---|
| **CanESM2** | 2.4 | 2.3 | 3.8 |
| **CanESM5** | 2.8 | 1.9 | 5.7 |

The Transient Climate Response (TCR) of the model is given by the temperature change in the 1pctCO2 experiment, averaged over the 20 years centered on the year of $CO_2$ doubling (year 70), relative to the piControl. For CanESM5 the TCR is 2.8 K, an increase of 0.4 K over that seen in CanESM2. The CanESM5 TCR is larger than seen in any CMIP5 models, and significantly higher than the CMIP5 mean value of 1.8 K (Flato et al., 2013). The likely range ($\rho > 0.66$) of TCR was given by the IPCC AR5 as 1.0-2.5 K (Collins et al., 2013), while more recent observational based estimates quote a 90% range of 1.2 to 2.4 K (Schurer and Hegerl, 2018), again subject to significant observational and methodological uncertainty.

The Transient Climate Response to Cumulative Emissions (TCRE), incorporates the transient climate sensitivity together with the carbon sensitivity of the system (Mathews et al., 2009). It is defined as the ratio of global mean surface warming to cumulative carbon emissions, over the 20 years centered on $CO_2$ doubling in the 1pctCO2 experiment, with units K EgC$^{-1}$. The metric is of major policy relevance, and is widely used to estimate the allowable emissions to reach given temperature targets. The TCRE of CanESM5 is 1.9 K EgC$^{-1}$, slightly lower than the CanESM2 value of 2.3 K EgC$^{-1}$. The reduction in TCRE occurs despite the fact that CanESM5 has a larger temperature response (TCR) than CanESM2. The reduction occurs owing to significantly larger uptake of $CO_2$ by the land biosphere in CanESM5 relative to CanESM2 in the 1pctCO2 experiment. Gillett et al. (2013) estimated the TCRE in 15 CMIP5 models to range from 0.8 to 2.4 K EgC$^{-1}$, and the IPCC AR5 likely range was assessed as 0.8 to 2.5 K EgC$^{-1}$.

The Equilibrium Climate Sensitivity (ECS) is defined as the amount of global mean surface warming resulting from a doubling of atmospheric $CO_2$, and a key measure of the sensitivity to external forcing. Given the long equilibration time of the climate system, it is common to estimate ECS from the relationship between surface temperature change and radiative forcing, over the course of the first 140 years of the abrupt-4xCO2 simulation (Gregory et al., 2004). For CanESM5, the ECS is 5.7 K, a significant increase over the value of 3.8 K in CanESM2. Like TCR, the CanESM5 ECS value is larger than seen in any CMIP5 models, and significantly higher than the CMIP5 mean value of 3.2 K (Flato et al., 2013). The likely range for ECS was given by the IPCC AR5 as 1.5 to 4.5 K (Collins et al., 2013). CanESM5 falls outside this range, although it is worth noting that there are significant uncertainties in observational constraints of ECS. We also note, as above, that ECS is an emergent property in CanESM5 - no model tuning was done on the response to forcing.






A detailed explanation of the reasons behind the increased ECS in CanESM2 over CanESM5 is beyond the scope of this paper.
However, the effective radiative forcing (Forster et al, 2016) in CanESM5 due to abrupt quadrupling of $CO_2$ is very similar to
that in CanESM2, suggesting that changes in feedbacks rather than forcings are the source of the higher ECS. Indications are
that the increase in ECS is associated with cloud and surface albedo feedbacks, with sea-ice likely playing an important role
in the latter effect. A more detailed examination of the changes in ECS due to cloud microphysics will be provided in a
companion paper in this special issue (Cole et al, 2019). The examination of climate change over the historical period in the
following section also reveals some further insights.

**6.2 Climate change over the historical period**

In this section we briefly discuss CanESM5 simulated changes in surface air temperature, sea-ice, and carbon cycle fluxes over

the historical period. We choose these as major emblematic variables of climate change.

**6.2.1 Surface temperature changes**

Global Mean Screen Temperature (GMST) changes in CanESM2 and CanESM5 are generally consistent with the observations

over the period from 1850 to around the end of the 20th century (Fig. 25a). However, from 2000 to 2014, the increase in GMST

is larger in the models than observed. Possible reasons for the divergence are i) forcing errors in the CMIP5 and/or CMIP6

forcing datasets, ii) natural internal variability, iii) incorrect partitioning of heat across components of the climate system or

iv) a higher climate sensitivity in the model than in the real world. The 25 realizations of CanESM5 (and 50 realization of

CanESM2) provide a good estimate the contribution of internal variability in the model. The observations fall outside the range

of this variability, and hence this cannot account entirely for the divergence between the model and observations (assuming

the model correctly captures the scale of internal variability). Trends computed from 1981 to 2014 show that the models are

warming at roughly twice the observed rate over this period (Fig. 25b). The spread across the 25 realizations from CanESM5

and 50 realizations from CanESM2 do not encompass the observations, reinforcing the point above. CanESM5 warms more

rapidly than CanESM2 on average, as would be expected from its higher ECS and TCR. There is however significant overlap

across the distribution of warming rates across the CanESM5 and CanESM2 ensembles. Interestingly, the lower tail of the

trend probability distribution functions aligns for the two models, but CanESM5 has a broader distribution, and a larger tail of

high warming realizations.



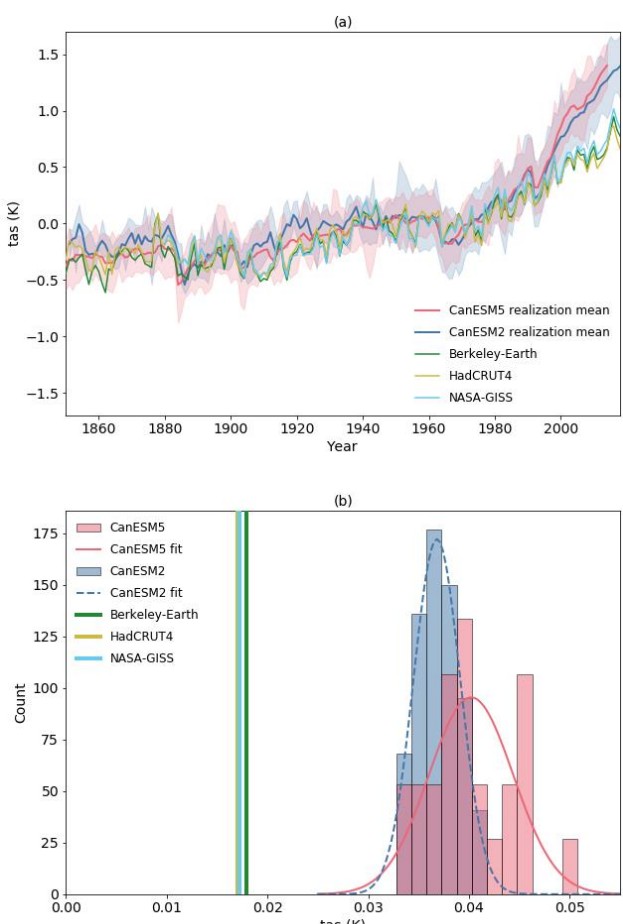

**Figure 25 (a) Global mean screen temperature in CanESM5, CanESM2 and various observational products and (b) histogram of historical trends over 1981 to 2014. In (a) the shaded envelopes represent the range over the CanESM2 50 member large ensemble and the CanESM5 25 member "p1" ensemble. In (b) fits of the normal distribution to the CanESM2 and CanESM5 distributions are also shown.**

The pattern of surface warming in CanESM5 over the historical period is shown in Fig 26a. The canonical features of global warming are consistent between the model and observations: greater warming over land than ocean, and Arctic amplified warming. The zonal-mean warming trends (Fig 26c) show that both CanESM2 and CanESM5 warmed more than the observations over most latitudes. Divergence between simulated and observed warming rates is largest in the high latitudes, notably over the Southern Ocean and north of 40°N. The larger warming in the CanESM5 ensemble mean, relative to the CanESM2 ensemble mean, largely occurs over the Arctic. However, there is a very large variability in Arctic warming trends in CanESM5, which most likely are responsible for the spread in GMST trends noted above. Some realizations have lower trends, which overlap with observed warming, while others exhibit considerably higher rates of Arctic warming. Observed





warming rates over the Arctic are also some of the most uncertain, due to data sparsity (HadCRUT is masked where
observations are not available).


**Figure 26: Surface temperature trends in CanESM5 (a), the difference in trend between CanESM5 and HadCRUT4**
**(b), and zonal mean of trends in CanESM2, CanESM5, and HADCRUT4 over 1981 to 2014 (c). The shaded envelopes**
**in (c) represent the range over the CanESM2 50 member large ensemble and the CanESM5 25 member "p1"**
**ensembles.**


### 6.2.2 Sea-ice changes

CanESM5 closely reproduces the observed reduction in Arctic September sea-ice extent (Fig. 27a). The trends from both the
50 CanESM2 ensemble members, and the 25 CanESM5 ensemble members, show a broad spread due to internal variability
(Fig. 27c). The observed trends lie close to the centre of the model distribution of trends. Given than CanESM5 warms more
rapidly than observed, the sea-ice sensitivity (rate of sea-ice decline normalized by the rate of warming) is likely too low
(Rosenblum and Eisenman, 2017; Winton, 2011).



In the Southern Hemisphere, observed annual mean Antarctic sea-ice extent showed a tendency to increase, before dramatic declines in the past few years (Fig. 27b). Both CanESM5 and CanESM2 show consistent declines over the historical period, with CanESM2 matching the climatological extent more closely. The spread of trends from the CanESM2 and CanESM5 ensembles suggest that the observed small positive trends in historical Antarctic sea-ice extent could plausibly have been due to internal climate variability (Fig. 27d).

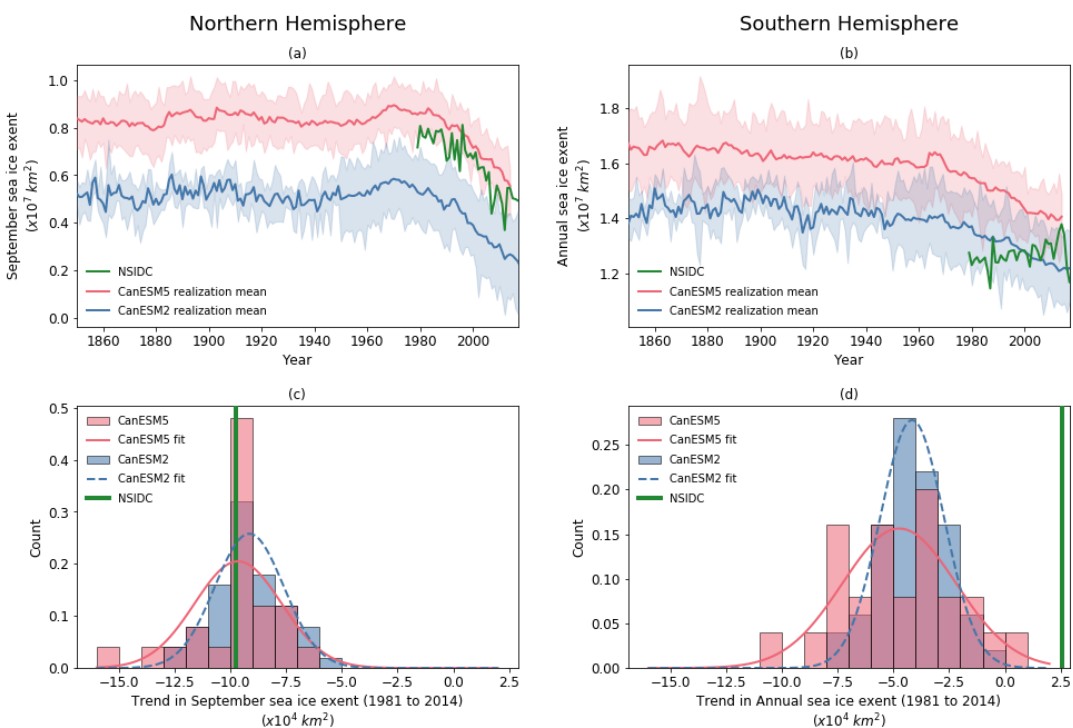

**Figure 27: Time series of sea-ice extent during (a) September in the Northern Hemisphere and (b) the annual mean in the Southern Hemisphere in CanESM5, CanESM2, and NSIDC satellite based observations. The histogram of trends over 1981 to 2014 in the lower panels. The shaded envelopes represent the range over the CanESM2 50 member large ensemble and the CanESM5 25 member "p1" ensemble. Fits of the normal distribution to the CanESM2 and CanESM5 histograms are also shown.**

**6.2.3 Historical carbon cycle changes**

The simulated global atmosphere-ocean ($F_O$) and atmosphere-land ($F_L$) $CO_2$ fluxes are shown if Fig. 28 for the historical period, along with their cumulative values over time. Also shown are the diagnosed anthropogenic fossil fuel emissions (E) that are consistent with the specified $CO_2$ pathway over the historical period, corrected for any drift in model's pre-industrial



control simulation (see Appendix F). The simulated values of $F_L$, $F_O$, and E are compared against estimates from the Global
Carbon Project (Le Quéré et al., 2018).

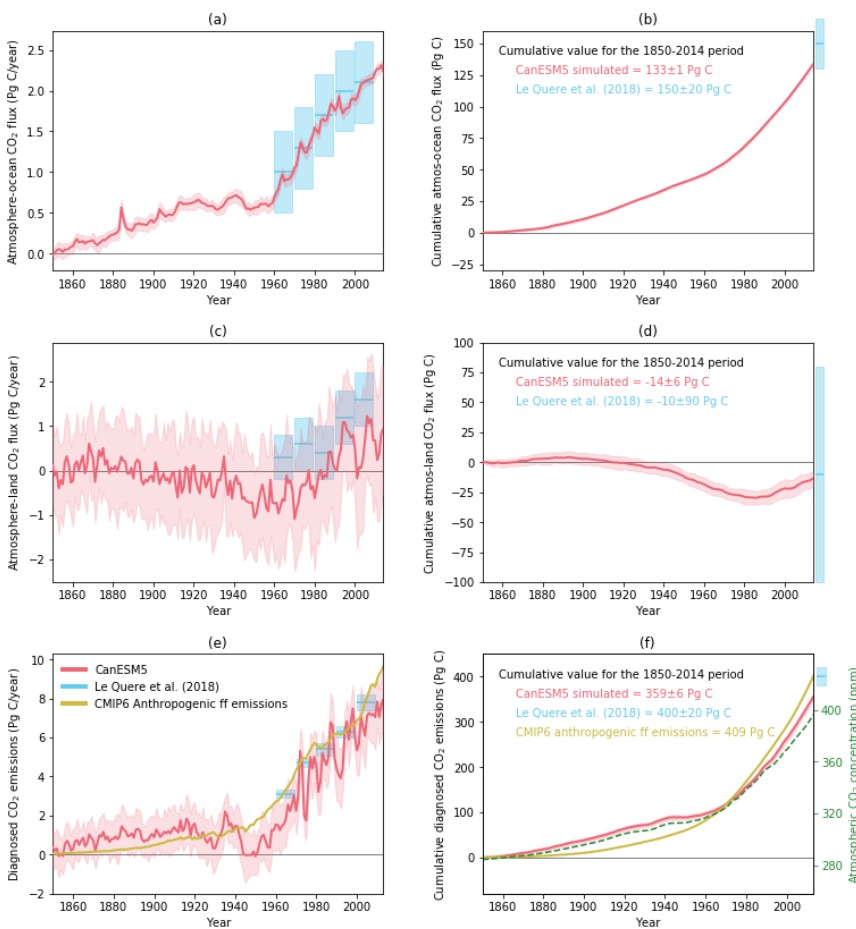


**Figure 28: Annual (left column) and cumulative (right column) global values of simulated atmosphere-ocean and**
**atmosphere-land CO2 fluxes, and diagnosed anthropogenic fossil fuel emissions, shown in blue colour. The model**
**values are shown as mean ± 1 standard deviation range and calculated based on the 25 ensemble members of the**
**historical simulation. Model values are compared against estimates from Le Quere et al. (2018).**


In Fig. 28a the simulated global atmosphere-ocean $CO_2$ fluxes compares reasonably well with observation-based estimates
from Le Quéré et al. (2018) for the decades of 1960s through to 2000s, although the simulated cumulative value of 133±1 Pg
C for the 1850-2014 period is on the lower end of the observation-based estimate of 150±20 Pg C (Fig. 28b). In contrast, the
simulated mean atmosphere-land $CO_2$ fluxes (Fig. 28c) are lower than their observation-based estimates for the decades of



1960s through to 2000s but their cumulative value of −14±6 Pg C over the 1850-2014 period compares well with the
observation-based estimate of −10±90 Pg C (Fig. 28d). The caveat here, of course, is the large uncertainty range in the
observation-based estimate of net cumulative atmosphere-land $CO_2$ flux (Appendix F). The reason the model's simulated
cumulative uptake of −14±6 Pg C over the 1850-2014 period compares well with the observation-based estimate of −10±90
Pg C, despite its weaker carbon sink since the 1960s (Fig. 28, panel c) is likely because the carbon source from land use change
emissions is also lower.

Panel e and f in Fig. 28 show the allowable diagnosed fossil fuel emissions and their cumulative values for the 1850-2014
period. The cumulative diagnosed fossil fuel emissions of 359±6 Pg C from the model for the period 1850-2014 are somewhat
lower than the CMIP6 and Le Quéré et al. (2018) estimates of 409 and 400±20 Pg C, respectively.
**7 Conclusions**
CanESM5 is the latest coupled model from the Canadian Centre for Climate Modelling and Analysis. Relative to its
predecessor, CanESM2, the model has new ocean, sea-ice and coupling components, and includes updates to the atmospheric
and land surface. The model produces a stable pre-industrial control climate, and notwithstanding some significant biases,
CanESM5 is able to reproduce many features of the historical climate. Objective global skill metrics show that CanESM5
improves the simulation of observed large scale climate patterns, relative to CanESM2, for most variables surveyed.  A notable
feature of CanESM5 is its high equilibrium climate sensitivity of 5.7 K, an emergent property of the updated physics described
above. This higher climate sensitivity appears to be driven by increased cloud and sea-ice albedo feedbacks in CanESM5. The
first major science application of CanESM5 is for CMIP6, with over 50, 000 years of CanESM5 simulation and more than 100
PB of data submitted to the publicly available CMIP6 archive. The model source code is also openly published for the first
time. Going forward CanESM5 will continue to be used for climate science applications in Canada.

**8 Code availability**
The full CanESM5 source code is publicly available at https://gitlab.com/cccma/canesm. The version of the code which can
be used to produce all the simulations submitted to CMIP6, and described in this manuscript, is tagged as *v5.0.3*, and has the
associated DOI: 10.5281/zenodo.3251113.

**9 Data availability**
All CanESM5 simulations conducted for CMIP6, including those described in this manuscript, are publicly available via the
Earth System Grid Federation (ESGF). All observational data used is publicly available. Data sources and citations are
provided in Appendix F.





**Appendices**

**Appendix A: Exchanges through the coupler**

**Table A1: Fields received by CanAM from CanCPL. The representative area may be the full AGCM grid cell (land, ocean, and ice), "C", open ocean, "O", sea-ice, "I", or the combination. Fields may be instantaneous, "Inst", or averaged over the coupling cycle, "avg".**

| Field Received | Field Description | Area | Avg |
|---|---|---|---|
| SICN_atm | sea ice fraction | OI | Inst |
| SIC_atm | ice water equivalent of sea ice | OI | Inst |
| SNO_atm | snow water equivalent over sea ice | I | Inst |
| GT_atm | sea surface temperature | O | Inst |
| CO2flx_atm | CO2 flux | OI | Inst |

**Table A2 Fields sent from CanNEMO to CanCPL. Descriptions as in Table A1.**

| Field Sent | Field Description | Area | Avg |
|---|---|---|---|
| OIceFrc | sea ice fraction | OI | Inst |
| OIceTck | ice water equivalent of sea ice | OI | Inst |
| OSnwTck | snow water equivalent over sea ice | OI | Inst |
| O_SSTSST | sea surface temperature | O | Inst |
| O_TepIce | sea ice surface temperature | I | Inst |
| O_CO2FLX | CO2 flux | OI | Inst |

**Table A3 Fields received by CanNEMO from CanCPL. Descriptions as in Table 1.**

| Field Received | Field Description | Area | Avg |
|---|---|---|---|
| O_OTaux1 | Atm-ocn wind stress (x) | O | avg |
| O_OTauy1 | Atm-ocn wind stress (y) | O | avg |
| O_ITaux1 | Atm-ice wind stress (x) | I | avg |
| O_ITauy1 | Atm-ice wind stress (y) | I | avg |
| O_QsrMix | solar heat flux mixed over ocean-ice | OI | avg |
| O_QsrIce | solar heat flux over sea ice | I | avg |
| O_QnsMix | non-solar heat flux mixed over ocean-ice | OI | avg |
| O_QnsIce | non-solar heat flux over sea ice | I | avg |
| OTotEvap | Total evaporation (evap + sublimation ) | OI | avg |
| OIceEvap | sublimation over sea ice | I | avg |
| OTotSnow | Snow | C | avg |





| OTotRain | Rain | C | avg |
|---|---|---|---|
| O_dQnsdT | non-Solar sensitivity to temperature | I | avg |
| O_Runoff | runoff | OI | avg |
| O_Wind10 | 10 meter wind | C | avg |
| O_TauMod | ocean wind stress modulus | O | avg |
| O_MSLP | Mean sea level pressure | C | avg |
| O_AtmCO2 | atm CO2 concentration | C | avg |


**Table A4 Fields sent from CanAM to CanCPL. Descriptions as in Table 1.**


| Field Sent | Field Description | Area | Avg |
|---|---|---|---|
| UFSO_atm | Atm-ocn wind stress (x) | O | avg |
| VFSO_atm | Atm-ocn wind stress (y) | O | avg |
| UFSI_atm | Atm-ice wind stress (x) | I | avg |
| VFSI_atm | Atm-ice wind stress (y) | I | avg |
| FSGO_atm | Solar heat flux over ocean | O | avg |
| FSGI_atm | Solar heat flux over ice | I | avg |
| BEGO_atm | Total heat flux over ocean | O | avg |
| BEGI_atm | Total heat flux over sea ice | I | avg |
| RAIN_atm | Total liquid precipitation | C | avg |
| SNOW_atm | Total solid precipitation | C | avg |
| BWGO_atm | ocean freshwater budget (P-E) | O | avg |
| BWGI_atm | sea ice fresh water budget | I | avg |
| SLIM_atm | non-Solar sensitivity to temperature | I | avg |
| RIVO_atm | River discharge | OI | avg |
| SWMX_atm | Mixed 10 meter wind | C | avg |
| PMSL_atm | Mean sea level pressure | C | avg |
| CO2_atm | Atm CO2 concentration | C | avg |









## Appendix B: Code management and model infrastructure

**Table B1: Code management**

| Item | Description |
|------|-------------|
| Source control | Each model component and supporting tools are version controlled in a dedicated git repository. Specific component versions are tracked as submodules by the CanESM super-repo, to define a version of CanESM. |
| Branching structure / workflow | Development of CanESM5 code follows a *gitflow* like workflow, commonly found in industry. Each logical unit of work is first described by an *issue*. Code changes are implemented on a dedicated feature branch. For simplicity, the feature branch is created in all submodules. Upon completion and acceptance, the feature branch is merged back onto the *develop_canesm* branch, which represents the latest state of the coupled model. Periodic tags on the *develop_canesm* branch mark stable versions of the model, which are then used for production purposes. The model version used for CMIP6 production is tagged as "CanESM.v5.0.0", and can be used to reproduce all existing CMIP6 simulations. A series of modified git commands is used to aid in working with submodules. |
| Versioning | Release versions of CanESM are tagged on the *develop_canesm* branch. Tags appear as CanESM.vXYZ, where X is the major version, Y is a minor number, and Z is a bugfix level number. For example, CanESM.v5.0.2. Over the course of CMIP6 development, only bit-pattern preserving changes have been accepted. |
| Forcing & initialization files | Forcing and initialization files are important for reproducibility, but not directly amenable to version control. An additional repository named *CanForce* contains the source code for scripts which produced the original input files. Input files are also checksummed, and a list of these checksums is tracked in the CanESM super-repository. |
| External dependencies | Specific versions of third party libraries, such as NetCDF, are loaded via an initialization procedure. Third party library source code is not directly tracked. |



**966**    **Table B2: Process for running CanESM**

| Item | Description |
|------|-------------|
| Run setup | Runs are setup on the ECCC HPC using a single entry point script (*setup-canesm*), which recursively clones the CanESM super-repository, and extracts some specific run configuration files. Hence, each run has a self-contained, full copy of the CanESM source code. This isolates runs from "external" changes, and also allows experimentation without affecting runs. When generating ensembles, code sharing between members is possible. *setup-canesm* also undertakes logging, recording which specific commit of CanESM was used in the run. |
| Run time environment | CanESM5 is run under Linux on ECCC's HPC. The user environment begins as only containing the path to *setup-canesm*. A machine-specific environment setup files is extracted from CCCma_tools by this utility script, and is sourced to define the runtime environment. The runtime environment essentially re-defines the PATH variable to point to the locally extracted scripting, as well as defining a host of machine-specific environment variables required at runtime. |
| Compilation | *setup-canesm* extracts utility compilation scripts. Ultimately, compilation scripts call the make utility to compile the code. The compilation of CanNEMO depends on the makenemo utility included in the source. Compilation of CanAM and CanCPL is done with makefiles, which are generated by the build-exe script, which determines required dependencies. |
| Configuration | CanESM runs are configured via the *canesm.cfg* file, which is extracted from the CanESM super-repo by *setup-canesm*. The configuration file allows selection of type of experiment (forcing files), start and end dates, diagnostics to be undertaken, and various options like dumping files to tape and deleting files. This configuration file is automatically captured in a dedicated configuration repository for posterity. |
| Sequencing | A legacy set of sequencing scripting is used to run CanESM simulations. In essence, a script called *cccjob* uses the information in *canesm.cfg* to create a sequential *string* of bash scripts, which run the model, compute diagnostics, and so forth. Such *jobstrings* are submitted to the HPC scheduler, and iterated over in sequence by a series of scripts contained in the *CCCma_tools* repository. |





| Strict checking | "Strict checking" is implemented during compilation, configuration, and during each increment over which the model is run when in production mode for official activities like CMIP6. Strict checking ensures that any source code changes have been committed, and that any configuration changes are captured in a dedicated repository. |
|---|---|

967

**Appendix C: Code optimization**

**Table C1: Description of optimization improvements to CanESM5. See Fig. 3 for a graphical representation.**

| Description of change | Throughput improvement (ypd) |
|---|---|
| Several I/O heavy operations, such as splitting and repacking files that were running in serial with the model execution were switched to run in parallel on the post-processing machine. In addition, the job submission scripting was simplified. | 0.4 |
| Splitting multi-year forcing files into yearly chunks resulted in speed improvement due to the non-sequential access of the CCCma file format. | 1.1 |
| Compiler flag optimization. Specifically the "-fp-model precise" was replaced with "-mp1" flag in the final 32-bit version, and the "-init=arrays -init=zero" flags were eliminated. The optimization level was increased from "-O1" to "-O2". | 1.5 |
| Adding a node to the CanAM component to speed up spectral transforms, and implement sharing of one node with the coupler (no increase in the overall node count). | 1.2 |
| Converting CanAM from 64 to 32 bit numerics | 4.1 |
| Writing model output from different cores/tasks into separate files (labeled in Fig. 3 as "parallel I/O"), and rebuilding them in parallel on the post-processing machine. | 0.4 |
| Changing model execution from occurring in monthly chunks (with re-initialization from restarts at the beginning of every month), to occurring in annual chunks. | 2.6 |



# Appendix D: CMIP6 MIP participation and model variants

**Table D1: List of MIPs and model variants of CanESM5 planned for submission to CMIP6.**

| MIP | Model variant |
|---|---|
| DECK-historical | CanESM5-p1, CanESM5-p2, CanESM5-CanOE-p2 |
| C4MIP | CanESM5-p1, CanESM5-p2 |
| CDRMIP | CanESM5-p1, CanESM5-p2 |
| CFMIP | CanESM5-p2 |
| DAMIP | CanESM5-p1 |
| DCPP | CanESM5-p2 |
| FAFMIP | CanESM5-p2 |
| GeoMIP | CanESM5-p2 |
| GMMIP | CanESM5-p2 |
| ISMIP6 | CanESM5-p1, CanESM5-p2 |
| LS3MIP | CanESM5-p1, CanESM5-p2 |
| LUMIP | CanESM5-p1, CanESM5-p2 |
| OMIP | CanESM5, CanESM5-CanOE (uncoupled). |
| PAMIP | CanESM5-p2 |
| RFMIP | CanESM5-p2 |
| ScenarioMIP | CanESM5-p1, CanESM5-p2, CanESM5-CanOE-p2 |
| VolMIP | CanESM5-p2 |
| CORDEX | N/A (CanRCM) |
| DynVar | CanESM5-p2 |
| SIMIP | CanESM5-p1, CanESM5-p2 |

972

**Appendix E: Comparison between p1 and p2**

Sections 2.5 and 3.4 described the technical differences between perturbed physics members p1 and p2, submitted to the CMIP6 archive. Here we provide a preliminary analysis of the differences between the two model variants.

Fig. E1 shows surface air temperature and precipitation averaged over 200 years of piControl experiment, for p1, p2 and the difference between them. Notable in the differences are the "cold" spots in Antarctica, which arise from a mis-specified land fraction in p1, and were resolved in p2. Otherwise there are no significant differences.

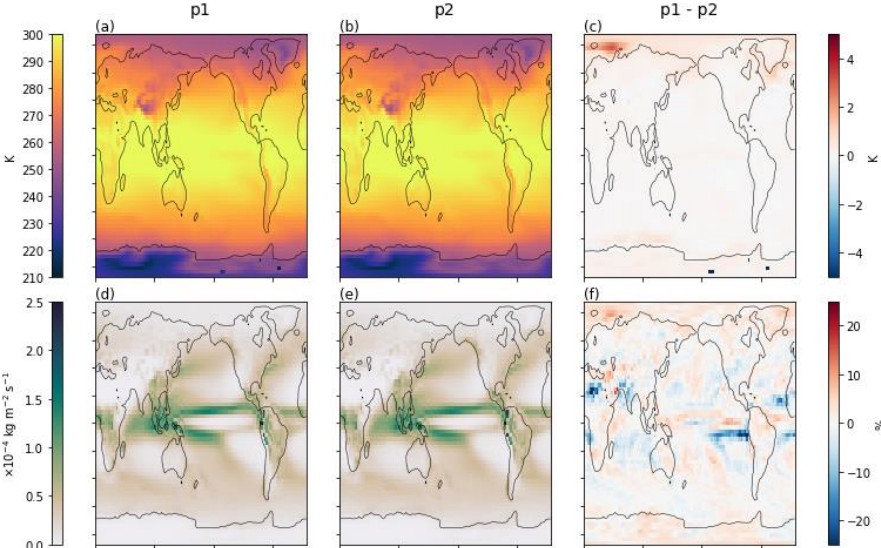

**Fig. E1: Climatologies of surface air temperature (a, b) and precipitation (d, e,) computed over 200 years of piControl simulation of the p1 (a, d) and p2 (b, e) model variants, and the differences between p1 and p2 (c, f).**

Fig E2 shows the ocean surface wind-stress. The blockiness of the field in p1 is evident, as a result of conservative remapping from CanAM. In p2, bilinear remapping was used and the field is smooth on the NEMO grid. The non-smooth nature of wind-stress in p1 resulted in, for example, banding in vertical ocean velocities at 100 m depth, as also shown in Fig. E2d. This does not occur in p2.


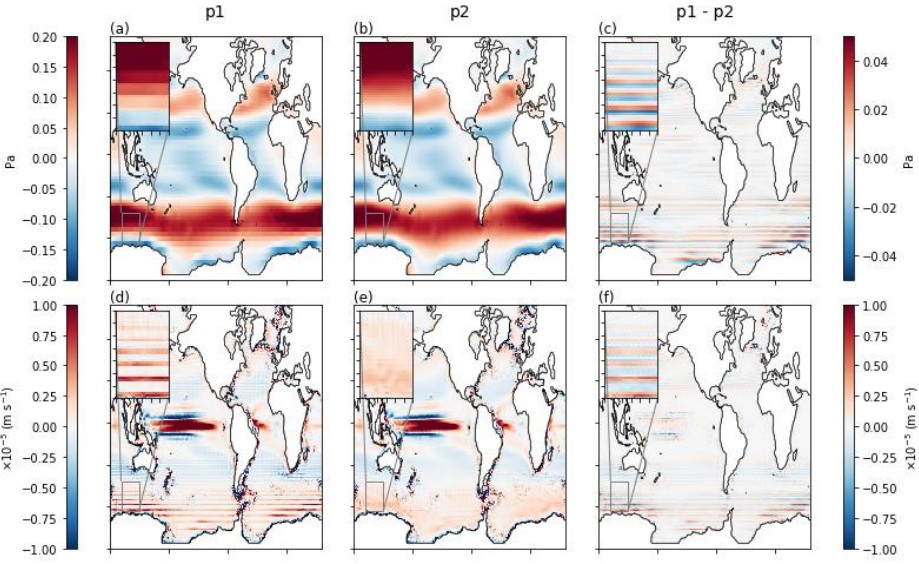


**Fig. E2: Climatologies of surface ocean zonal wind stress (a, b) and vertical velocity near 100 m depth (d, e) computed**
**over 200 years of piControl simulation of the p1(a, d) and p2 (b, e) model variants, and the differences between p1**
**and p2 (c, f). Results are shown on the native NEMO grid. The insets show an enlargement of the Southern Ocean**
**south of Australia.**

The response to CO2 forcing in the 1pctCO2 experiments in p1 and p2 is shown in Fig. E3. The global mean top of atmosphere
radiation (Fig E3a) and surface air temperature (Fig E3b) responses are indistinguishable, and hence the TCR of these model
variants is the same. The ocean is cooler, on average, in p2, but the perturbative response in p1 and p2 are similar (Fig E3c).
Ocean surface CO2 flux is also statistically indistinguishable between the variants (Fig E3d).

Maps of the perturbative response, computed as the mean over the 20 years centered on $CO_2$ doubling in the 1pctCO2
experiments, minus the piControl, are shown in Figs. E4 and E5. There are no fundamental differences in the surface climate
response between the two model variants.









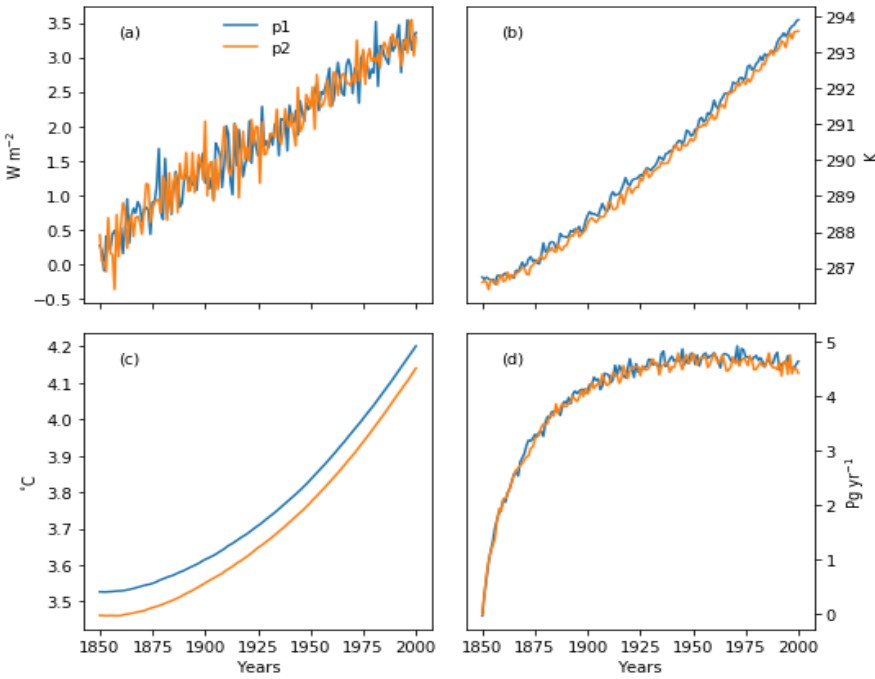


**Fig. E3: Global averages of (a) top of atmosphere net radiative flux, (b) surface air temperature, (c) volume averaged**


**ocean temperature and (d) surface ocean CO2 flux in the 1pctCO2 simulations from the p1 and p2 model variants.**



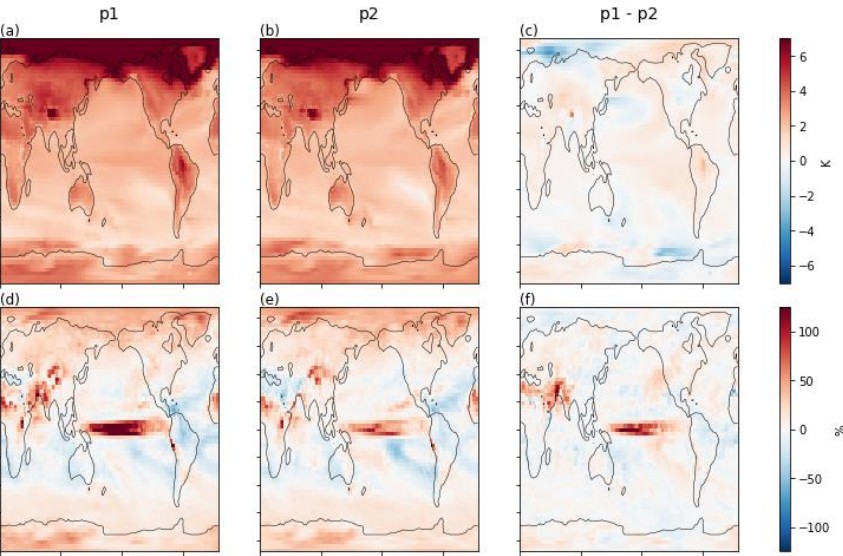


**Fig. E4: Perturbation of surface air temperature (a, b) and precipitation (d, e) computed as the mean over the 20**


**years centered on CO₂ doubling in the 1pctCO2 experiment, minus the mean from 200 years of piControl simulation**


**of the p1 (a, d) and p2 (b, e) model variants, and the differences between p1 and p2 (c, f).**



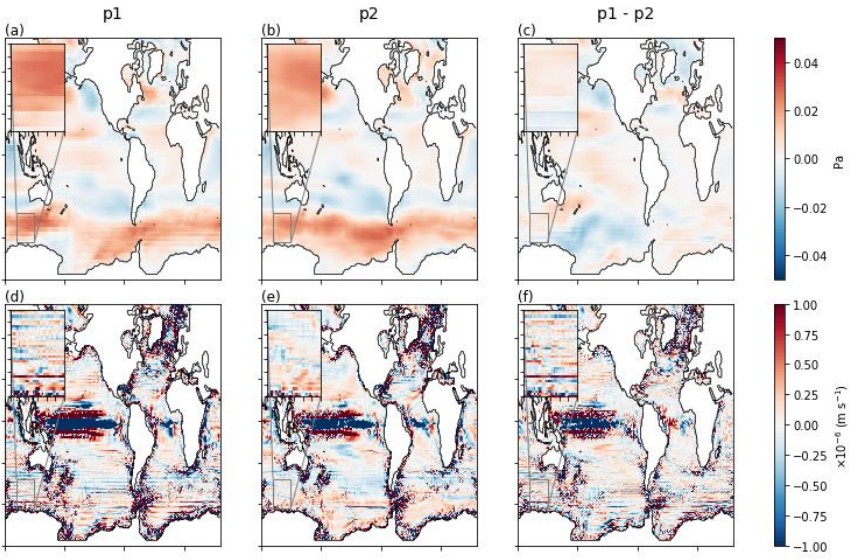


**Fig. E5: Perturbations of surface ocean zonal wind stress (a, b) and vertical velocity near 100 m depth (d, e)**
**computed as the mean over the 20 years centered on $CO_2$ doubling in the 1pctCO2 experiment, minus the mean over**
**200 years of piControl simulation of the p1 (a, d) and p2 (b, e) model variants, and the differences between p1 and p2**
**(c, f). Results are shown on the native NEMO grid. The insets show an enlargement of the Southern Ocean south of**
**Australia.**















**Appendix F: Data sources, variables, and derived quantities**

**Table F1: List of figures, CanESM5 CMIP6 variables, and observations used, and the time-periods of analysis.**

| Fig No | CMIP6 variables | CMIP6 experiment and (variant label) | Observations | Time period of analysis |
|---|---|---|---|---|
| 4 | tas | piControl, historical, abrupt-4xCO2, 1pctCO2, SSP5-85, SSP (r1i1p1f1) | n/a | 1850-2100 |
| 5 | rtmt, hfds, thetao, tas, wfo, zos, sivol, snw, nep, fgco2, cLand, dissic (piControl) | piControl (r1i1p1f1) | n/a | 5200-6200 |
| 6 | As labelled. | historical (r1i1p1f1) | ERA5, GPCP, GBAF, WOA09, AVISO, GLODAPv2.2016, ISCCP-H, AVISO MDT, NSIDC, Landschützer et al. (2015) | |
| 7 | tas, pr, psl | historical (r1i1p1f1) | ERA5, GPCP | 1981-2010 |
| 8 | clt | historical (r1i1p1f1) | ISCCP-H | 1991-2010 (ISCCP data period) |
| 9 | ta | historical (r1i1p1f1) | ERA5 | 1981-2010 |
| 10 | ua | historical (r1i1p1f1) | ERA5 | 1981-2010 |
| 11 | uas | historical (r1i1p1f1) | ERA5 | 1981-2010 |
| 12 | gpp, hfls, and hfss | historical (r1i1p1f1) | GBAF | 1982-2009 (GBAF data period) |
| 13 | gpp, hfls, and hfss | historical (r1i1p1f1) | GBAF | 1982-2009 (GBAF data period) |





| 14 | gpp, tas, pr | historical (r1i1p1f1) | GBAF | 1982-2009 (GBAF data period) |
|---|---|---|---|---|
| 15 | tos, sos, zos | historical (r1i1p1f1) | WOA09, AVISO MDT | 1981-2010 |
| 16 | thetao, so | historical (r1i1p1f1) | WOA09 | 1981-2010 |
| 17 | msftmz | historical (r1i1p1f1) | - | 1981-2010 |
| 18 | hfbasin | historical (r1i1p1f1) | Ganachaud and Wunsch (2003) | 1981-2010 |
| 19 | siconc, sithick | historical (r1i1p1f1 to r25i1p1f1) | NSIDC | 1981-2010 |
| 20 | siconc | historical (r1i1p1f1) | NSIDC | 1981-2010 |
| 21 | dissic, no3, o2 | historical (r1i1p1f1) | GLODAPv2.2016 | 1981-2010 |
| 22 | fgco2 | historical (r1i1p1f1) | Landschützer et al. (2015) | 1982-2010 (Landschützer data period) |
| 23 | tos | historical (r1i1p1f1 to r10i1p1f1) | HadISST | 1850-2014 |
| 24 | psl | historical (r1i1p1f1) | ERA5 | 1981-2010 |
| 25 | tas | historical (r1i1p1f1 to r25i1p1f1) | Berkeley-Earth, HadCRUT4, NASA-GISS | Time series: 1850-2014 Trends: 1981-2014 |
| 26 | tas | historical (r1i1p1f1 to r25i1p1f1) | Berkeley-Earth, HadCRUT4, NASA-GISS | 1981-2014 |
| 27 | siconc | historical (r1i1p1f1 to r25i1p1f1) | NSIDC | Time series: 1850-2014 Trends: 1981-2014 |
| 28 | fgco2, nep, co2 | historical (r1i1p1f1 to r25i1p1f1) | Le Quéré et al. (2018) | 1850-2014 |









**Table F2: List of observational products used.**

| Data source | Citation |
|---|---|
| AVISO MDT | https://www.aviso.altimetry.fr/en/data/products/auxiliary-products/mdt.html |
| ERA5 | Copernicus Climate Change Service (2017) |
| GPCP | Adler et al. (2017) |
| ISCCP-H | Young et al. (2018); Rossow et al. (2016) |
| GBAF | Jung et al. (2009) |
| World Ocean Atlas 2009 (WOA09) | Locarnini et al. (2009); Antonov et al. (2010) |
| NSIDC sea-ice concentration | Peng et al. (2013);  Meier et al. (2017) |
| PIOMAS | Zhang et al. (2003) |
| GIOMAS | Zhang et al. (2003) |
| GLODAPv2 | Lauvset et al. (2016) |
| Landschützer | Landschützer et al. (2015) |
| HadISST | Rayner et al. (2003) |
| Berkley Earth | http://berkeleyearth.org/land-and-ocean-data/ |
| HadCRUT4 | Morice et al. (2012) |
| NASA-GISS | GISSTEMP Team (2019); Lenssen et al. (2019) |
| Global Carbon Budget 2018 | Le Quéré et al. (2018) |



In Figure 28 the diagnosed allowable anthropogenic fossil fuel emissions are calculated via Equation (F1):
$$\frac{d\,[CO2]}{dt} = E - F_L - F_O = E - (F'_L - E_{LUC}) - F_O \qquad (F1).$$

In these historical simulations, the concentration of atmospheric $CO_2$ is specified (that is the term $d[CO_2]/dt$ is known) and the
model's land and ocean carbon cycle components simulate atmosphere-land ($F_L$) and atmosphere-ocean ($F_O$) $CO_2$ fluxes,
respectively. The $F_L=F'_L-E_{LUC}$ term includes natural atmosphere-land $CO_2$ flux ($F'_L$) and the emissions associated with land
use change ($E_{LUC}$) which are calculated interactively in the model in response to the historical increase in cropland area. As a
result, the term E can be calculated and represents the allowable anthropogenic fossil fuel emissions.

Le Quéré et al. (2018) do not provide a direct value of net cumulative atmosphere-land CO2 flux ($F_L$). Instead, they separately
provide estimates of cumulative values of $F'_L$ (185±50 Pg C) and $E_{LUC}$ (195±75) in their Table 8. Here, we calculate
observation-based value of $F_L=F'_L-E_{LUC} =185−195 = −10$ Pg C and its uncertainty as 90 Pg C (the uncertainty is calculated as,
$\sqrt{(50^2 + 75^2)} = 90.13$ PgC. The large uncertainty range for the observation-based estimate of cumulative $F_L$ is therefore due
to large uncertainties in both land use change emissions and the natural atmosphere-land $CO_2$ flux.



**Author contributions**

NCS co-led CanESM5 development, contributed to CanNEMO and CMOC development and the data request, performed simulations, led the creation of the figures and wrote most of the manuscript; JNSC contributed to development of CanAM5, CanCPL and tuning of CanESM5, wrote the CanAM5 section, performed simulations and contributed to the data request; VK contributed to the development of CanAM, notably optimization, contributed to the data request, and performed production simulations; ML contributed to the development of CanAM, CanCPL and the data request; JS co-led CanESM5 development; NG contributed to CanESM5 development and tuning; JA contributed to CanCPL development, the data request, and led publication of data on the ESGF; VA contributed to the development of CLASS and CTEM; JC developed CanOE and contributed to CMOC development; SH produced many of the figures; YJ contributed to the data request and conversion; WL contributed to CanNEMO development and ran simulations; FM contributed to the CanESM5 software infrastructure; OS led ocean physics testing; provided a specific analysis that motivated the p2 variant; ChS contributed analysis of the land component; ClS contributed to CanESM5 software infrastructure; AS created Fig. 23, contributed to CanESM5 development, and performed simulations; LS developed CanCPL; KVS led the development and tuning of atmospheric model parameterizations; DY contributed to ocean model development, ocean and sea ice diagnostics for CMIP6, and performed production simulations; BW processed forcing datasets for CanAM; All authors contributed to editing the manuscript.

**Competing interests**

No competing interests.

**Disclaimer**

CanESM has been customized to run on the ECCC high performance computer, and a significant fraction of the software infrastructure used to run the model is specific to the individual machines and architecture. While we publicly provide the code, we cannot provide any support for migrating the model to different machines or architectures.

**Special issue statement (will be included by Copernicus)**

To be included in the CanESM5 special issue.

**Acknowledgements**

We acknowledge Dr. Michael Sigmond, Dr. Greg Flato and Dr. William Merryfield for helpful comments on a draft of the paper. CanESM5 was the cumulative result of work by many individuals, who we thank for their contributions. CanESM5 simulations were performed on ECCC's HPC, and CanESM5 data is served via the Earth System Grid Federation.



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
