# Peer review of "The Canadian Earth System Model version 5 (CanESM5.0.3)"

_Geoscientific Model Development, 2019_

## Referee Comment (RC1) · Anonymous Referee #1 · 13 Aug 2019

The paper describes the Earth-system model CanESM5, which has performed simulations in support of CMIP6. The paper itself is very well written with very few grammatical or typographical errors.

I was particularly impressed with the detailed discussion of how the model is managed with regard to the structure of the various git repositories. The improvements to model throughput are also impressive, going from 4.5 to 16 simulated years per day from the various changes described.

The model itself represents some quantities well and others not so well, as might be expected in any ESM. Given the model has a relatively coarse resolution and does not represent all processes the behaviour is not unreasonable.

[Figure]

I will structure my comments with some more major ones first followed by a small number of minor errors. I believe that with some small changes to the manuscript the paper would be ready for publication.

**Major Comments**

In the Introduction the authors briefly discuss CanESM5 within the modelling system of CCCma, which includes CanRCM, CMAM, and CanSIPS. It might be helpful to describe how these are connected - are they all entirely separate, or do they share components/models?

Figure 1 - I find this figure a little frustrating. While it is clear in naming the different component models that make up CanESM5, it does not describe how they are connected or each model's complexity. A reader must consult Appendix A, which is somewhat technical and is still not complete for all linkages as this only discusses what goes through the coupler. As it is this figure doesn't really show the "evolution of components" as it does not describe how complex each component model is in either CanESM2 or CanESM5, although it does show the changes to model version numbers or the change of model used.

What I felt was missing in section 2 was a description of just how complex the atmosphere model CanAM5 is. The description is very brief and readers are directed to Cole et al. (2019), which although will also be in the same special issue as this paper, is currently still in preparation. Also, this paper describing CanESM5 should stand enough on its own such that important details regarding component models should be presented here. I suggest that the authors expand section 2.1 enough to briefly cover the complexity of the atmosphere model and what processes (or not) are represented. There is also no mention of atmospheric chemistry or the complexity of the aerosol scheme.

The authors should ensure that each component model is described in sufficient detail to understand what processes it can simulate and how these models interact together

in the ESM. As this is an Earth-system model, I am interested in seeing just how the component models interact. The tables in Appendix A only cover what goes through the coupler, so methane emissions that are discussed on lines 125-130 aren't mentioned for instance - are there any others? Here what do the methane emissions do, is there a simple methane oxidation scheme, does it produce water vapour etc.?

At line 422 the authors mention "five realisations of CanESM2", and then later discuss a 50 member large ensemble of CanESM2. Is it just that only 5 members were submitted to CMIP5? These are first mentioned on line 844. Is there a reference or more detail that can be provided?

**Minor Comments**

Out of interest, why does the numbering system go from CanESM2 to CanESM5?

line 33 - I believe that "earth" should be capitalised in this context.

line 149 - I'm not sure why "converted into ice" is in quotes here. Is there a reason for this?

Section 2.3 - the way numbers and units are represented here are slightly different from other sections. The $\times$ symbol is rather large (compare to line 236), and units are given as $m^2/s$ rather than $m^2s^{-1}$ as is done elsewhere in the manuscript.

line 280 - why are "bilinear" and "conservative" in quotes? I don't think this is necessary here.

Section 5.4 - two different ways of doing a $\pm$ symbol are used (lines 621 and 642). Given the differences in Section 2.3 (mention above) I suggest that the authors double-check how numbers and symbols are presented for consistency.

Table B1 (versioning) - should this be "CanESM.vX.Y.Z"?

[Figure]

2019.

---

## Referee Comment (RC2) · Anonymous Referee #2 · 20 Aug 2019

Review of Swart et al. 2019: The Canadian Earth System Model version 5 (CanESM5).

General Comments:

This paper gives a clear and detailed description of the new Canadian Earth System model (CanESM5). The paper nicely describes the main developments relative to earlier Canadian models (e.g. CanESM2) and also provides a good overview of the primary model performance, sensitivities and associated biases, touching on (i) pre-industrial coupled climate stability, (ii) performance over the recent historical past and (iii) first-order sensitivity to increasing $CO_2$ concentrations. It will be a useful reference to users of the model and to climate analysts using CanESM5 CMIP6 results. I recommend the paper be published, conditional on one relatively "major" comment and a number of more "minor" comments being addressed. These are detailed below.

[Figure]

Major Comment:

The paper assesses the performance of CanESM5 over the latter part of the CMIP6 historical simulation period, using a number of standard diagnostics where CanESM5 is compared to observations. A number of model biases/shortcomings are identified, but the likely/probable cause of these biases is never discussed. While I recognise it is not straightforward to fully identify the underlying cause of coupled model errors, it would be a useful addition to the paper if, at least for the most important biases (e.g. excessive surface temperature trends post-1980, precipitation and cloud biases, excess heat uptake in the tropical ocean thermocline etc) some discussion of the cause of these biases was provided. This does not have to be exhaustive and can point to a deeper analysis and discussion to come elsewhere, but some pointers to the main causes of key biases would give the paper a bit more "meat". This comment also applies to the significant change (increase) in TCR and ECS in CanESM5 relative to CanESM2, a somewhat greater discussion on what is considered the leading cause(s) for this increase seems appropriate.

Minor comments:

Minor comments are listed with reference to the line numbers in the paper. Where I think some additional explanation would improve the paper I outline this within each point.

1. L50: "CanESM5 represents a major update to CanESM2": The reader is left wondering what happened to CanESM3 and 4 ?? Is there some reason naming jumped from "2" to "5"?

2. L86-L87: "the emission of mineral dust and DMS was improved...". It would be useful to know the degree to which dust, DMS and other natural (aerosol) emissions are fully prognostic in the model, allowing potential future feedbacks to be simulated, versus these using observation-based (and therefore time-invariant) prescribed input fields.

3. L101-L102: "The introduction of dynamic wetlands and their methane emissions is a new biogeochemical process...". Is methane a prognostic atmospheric variable in CanESM5? e.g. how do the wetland methane emissions influence the model radiation? There is no mention of interactive chemistry in CanESM5, I therefore assume methane is not a prognostic variable and that there are no internal (model) feedbacks between wetland methane emissions and climate. How methane (and other non-CO2 radiatively important gases e.g. ozone) are treated in CanESM5 needs to be more clearly explained.

4. L121: Why does CLASS use 4 PFTs but CTEM use 9? And is vegetation type/amount dynamically predicted in CanESM5 or (externally) prescribed? Please state what is done for the latter and explain the rationale for the former.

5. L304-306: It sound as though the primary tuning for the coupled pre-industrial model simulation was global mean surface temperature. Is this correct? It would seem to me that (a zero) global mean net top of atmosphere radiation would be a more appropriate target. The resulting net TOA balance looks to fairly close to zero (0.11Wm-2, figure 5a). Was the TOA net radiation budget also used as a tuning target?

6. L318-319: "The final adjustment was to the carbon uptake over land.......achieved via the parameter which controls the strength of the CO2 fertilization effect". Does this indicate some structural problems with the parameterization of CO2 fertilization? I ask because in L813-815 there is mention of; "significantly larger uptake of CO2 by the land biosphere in CanESM5 relative to CanESM2 in the 1pctCO2 experiment". Is this change in land uptake influenced by the tuning applied to the historical period CO2 fertilization term? Allied to this does CanESM5 include any representation of nutrient limitation of the land CO2 fertilization term?

7. L320: "Critically, no tuning was undertaken on the climate system response to forcing". This is not strictly true as the authors say that they tuned the model to improve aspects of the historical simulation, which does include perturbed CO2 forcing.

8. Figure 4: This figure appears on page 15 but is not discussed until much later. In itself that is okay, but then it might be worth adding a statement where the figure is first shown, saying why this figure is sufficiently important to be shown at the start of this section and that further discussion of the figure results appear in sections xx or yy.

9. Figure 5 (caption). Some of the figures are explicitly listed as being "global" or "global mean" while others are not. Those not classified as "global" or "global mean", if they are not, for what spatial region are they representative? If they are all global quantities then make this clear in the figure caption.

10. Figure 6 (caption): "Variables are labelled according to the names in the CMIP6 data request" As not all readers will be CMIP6 experts, it could be useful to list what this shorthand list means in real word descriptions, possibly in a table in an appendix.

11. L520-521: Some of the precipitation biases, while large in percentage terms, are actually very small in absolute terms (e.g. the subtropical East Pacific). Would it be better to plot absolute precipitation biases? as done for temperature and SLP.

12. For Figure 7 (and possibly 8). I suggest showing DJF and JJA seasonal means rather than annual means. This will help focus on biases and reduce the risk of error cancellation within the annual mean, e.g. winter vs summer surface temperature biases, better focus on summer vs winter monsoon precipitation biases etc.

13. Figure 8 and L530: It would be useful to plot TOA solar and longwave radiation fields and biases (e.g. vs CERES EBAF) alongside the cloud plots, given that the number 1 consequence of cloud errors are in their impact on radiation.

14. L589-591: The wording in these lines does not make sense.

15. Figure 13: (caption) "The shading presents the corresponding inter-quartile range": What range are we referring to here? For the model results is the range across ensemble members? What is the range with respect to the observations?

16. Figure 14: What occurrence of surface temperature and precipitation is plotted?

Daily mean values, monthly means ? Please state what is plotted.

17. L615-616: "Negative annual mean SSS biases occur under the region associated with excessive March sea-ice. . ." This seems counter to what I would expect. Generally with sea-ice formation there is brine rejection into the surface water, hence with excessive sea ice I would expect excessive brine rejection and a positive bias in SSS.

18. L676-678: Please provide references to PIOMAS and GIOMAS.

19. Figure 20: The excessive model sea-ice in the Labrador sea, does this have an impact on deep water formation and potentially help explain the weak (∼12Sv) AMOC in CanESM5?

20. Figure 20 needs a clearer figure caption and explanation. It is not clear whether the plot shows ocean to atmosphere or atmosphere to ocean C flux. I am guessing positive values in the North Atlantic indicate ingassing (e.g. atmosphere to ocean flux of C). The implied direction of the flux with respect to positive and negative values needs to be made clearer. Also figure b looks like the difference in flux CanESM5 minus Landschutzer, the figure caption suggests it is an absolute value of Landschutzer.

21. L773-774: Why use this definition of the Northern Annular Mode? Is this a standard circulation measure? Would it not be better to plot the NAO for the Northern hemisphere?

22. Figure 24 (caption) : "The colour scale is arbitrary" What does this mean? Presumably the scales are the same across the 4 figures ?

23. L813-816: The TCR and ECS both increase (by ∼16% and 50%) in CanESM5 relative to CanESM, yet the TCRE decreases by ∼16%. This is unusual. It is explained as being due to a "significantly larger uptake of $CO_2$ by the land biosphere in CanESM5 relative to CanESM2". Given the stated importance of TCRE for policymakers, are the authors confident that the increased $CO_2$ uptake is due to realistic and understood reasons?

24. L821-825: "ECS IS 5.7K": Actually, in figure 4 the global mean surface temperature does not look to have sufficiently stabilized by year 140 to derive an ECS value (e.g. it increase by ∼0.75C over years 90 to 150). Have the authors tried plotting the so-called Gregory linear regression of nett TOA radiation versus surface temperature to see whether there is a single or multiple linear relationships between these two, or whether a non-linear behaviour is evident?

25. Figure 26c: It is difficult to see any differences. This might be better plotted as a model to observation difference or ratio.

---

## Referee Comment (RC3) · Anonymous Referee #3 · 21 Aug 2019

Review of The Canadian Earth System Model version 5 (CanESM5.0.3) by Neil et al. for GMD.

The paper describes the last version of CanESM. The main goal is to provide a reference for people who will analyse CMIP6 model outputs.

Like all model description paper, there is a conflict between the need to be as comprehensive as possible, while keeping a reasonable paper length. In this point of view, the paper achieves a good and relevant compromise. It includes a short description of the model components, with all references for the reader who wants to go further in details. It contains a classical choice of model diagnostic to evaluate the model climate. It contains also informations0, of the model sensitivity to standardised scenarios of $CO_2$ increase. This content nicely matches the main objective of the paper.

I really appreciated the honesty of the authors, particularly when they described the consequence of the bug in p1 version, corrected in the p2 version.

The paper also includes information about model quality control and performance. This is not of interest for CMIP6 data users. But model developers will find useful information on the way other teams work. Information that is hardly presented elsewhere.

The general organisation of the paper is good. As non-native English speaker, I won't comment the quality of the syntax. I found very few typos, and thanks the authors for their careful proofreading.

The main weakness of the paper is that the model is assessed in comparison with data and with the previous version of the model. There is no comparison with the CMIP5 models. This is not a major concern, as the model outputs will be used during CMIP6, and CamESM will be compared to CMIP5 and CMIP6 model in the next months by other authors.

To sum up, this is a very good paper that perfectly fits the reason why GMD has been created for. I think it can be published after a few technical corrections.

Major concern

Line 351. I do not understand how "version control, run isolation, strict checking and logging" can insure that the climate is reproducible. I agree that up to now, nobody has observed that lack of bit identical reproducibility in an ESM can drive to a different climate. But we have also theories of deterministic chaos showing that this is possible.

Minor concerns

Line 125 and following. Is there some specific representation of the urbanised areas ?

Line 151 and following. The melt water of the glaciers goes to the runoff scheme. How did you design a 'river' routing scheme for the ice sheets ? (from slopes ?)

Line 162. For the NEMO TKE scheme, a better reference than Gaspar et al. 1990

could be Blanke, B. and Delecluse, P.: Variability of the Tropical Atlantic Ocean simulated by a general circulation model with two mixed layer physics, Journal of Physical Oceanography, 23, 1363–1388, 1993.

Line 218. What is the computing coast of CMOC and CanOE compared to NEMO dynamics and to LIM2 ? It is significant or not ?

Line 777. Bentson et al. 2013 is not in the bibliography. Probably a typo, for Bentsen.

Line 809. Mathews et al. 2009 is not in the bibliography. Probably a typo, for Matthews.

Line 840. What is "Global Mean Screen Temperature" ? GMST generally stands for Global Mean Surface Temperature.

Line 1335. Year of the paper is embedded in the URL of the DOI.
* * *

---

## Author Comment (AC1) · 21 Sep 2019

Review comments in black.

Responses in green.

The paper describes the Earth-system model CanESM5, which has performed simulations in support of CMIP6. The paper itself is very well written with very few grammatical or typographical errors. I was particularly impressed with the detailed discussion of how the model is managed with regard to the structure of the various git repositories. The improvements to model throughput are also impressive, going from 4.5 to 16 simulated years per day from the various changes described. The model itself represents some quantities well and others not so well, as might be expected in any ESM. Given the model has a relatively coarse resolution and does not represent all processes the behaviour is not unreasonable.

We thank the reviewer for their constructive comments.

**Major Comments**

In the Introduction the authors briefly discuss CanESM5 within the modelling system of CCCma, which includes CanRCM, CMAM, and CanSIPS. It might be helpful to describe how these are connected - are they all entirely separate, or do they share components/models?

To clarify the connection we have updated the sentence to read "CanESM, forms the basis of the CCCma modelling system, and shares components with the Canadian Regional Climate Model (CanRCM) for finer scale modelling of the atmosphere (Scinocca et al., 2016), the Canadian Middle Atmosphere Model (CMAM) with atmospheric chemistry (Scinocca et al., 2008), and the Canadian Seasonal to Interseasonal Prediction System which is used for seasonal prediction and decadal forecasts (CanSIPS, Merryfield et al., 2013)."

Figure 1 - I find this figure a little frustrating. While it is clear in naming the different component models that make up CanESM5, it does not describe how they are connected or each model's complexity. A reader must consult Appendix A, which is somewhat technical and is still not complete for all linkages as this only discusses what goes through the coupler. As it is this figure doesn't really show the "evolution of components" as it does not describe how complex each component model is in either CanESM2 or CanESM5, although it does show the changes to model version numbers or the change of model used.

We agree this figure was not adding much value, and have removed it in the revision. We have expanded the description of the model in Sections 2.1 and 2.6, providing the reader with more clarity on the complexity of each model component.

What I felt was missing in section 2 was a description of just how complex the atmosphere model CanAM5 is. The description is very brief and readers are directed to Cole et al. (2019), which although will also be in the same special issue as this paper, is currently still in preparation. Also, this paper describing CanESM5 should stand enough on its own such that important details regarding component models should be presented here. I suggest that the authors expand section 2.1 enough to briefly cover the complexity of the atmosphere model and what processes (or not) are represented. There is also no mention of atmospheric chemistry or the complexity of the aerosol scheme.

We have expanded section 2.1 to provide a more complete overview of the complexity of CanAM, including the treatment of aerosols and atmospheric chemistry. We have also added a new section (2.6) to describe the treatment of greenhouse gasses across the ESM components.

Added to Section 2.1.:

"Version 5 of the Canadian Atmospheric Model (CanAM5) employs a spectral dynamical core with a hybrid sigma-pressure coordinate in the vertical.  The package of physical parameterizations used by CanAM5 are based on an updated version of its predecessor, CanAM4 (von Salzen et al., 2013).  The physics package includes a prognostic cloud microphysics scheme governing water vapour, cloud liquid water, and cloud ice; a statistical layer-cloud scheme; and independent cloud-base mass-flux schemes for both deep and shallow convection. Aerosols are parameterized using a prognostic scheme for bulk concentrations of natural and anthropogenic aerosols, including sulfate, black and organic carbon, sea salt, and mineral dust; parameterizations for emissions, transport, gas-phase and aqueous-phase chemistry, and dry and wet deposition account for interactions with simulated meteorology. CanAM5 employs a triangular truncation at total wavenumber 63 (T63) corresponding to an approximate isotropic resolution of 2.8 degrees in both latitude and longitude. In the vertical, 49 levels are employed with layer thicknesses that increase monotonically from approximately 100 m at the surface to 2km at ~1hPa – the domain lid."

**"2.6 Treatment of greenhouse gases**

CanESM5 represents radiative forcing from individual greenhouse gases (GHGs). Aside from $CO_2$, the concentrations of all radiatively active gases are specified and transiently evolve.  Of these, $CH_4$, $N_2O$, and families of CFCs are assumed to be well-mixed, while, $O_3$ is specified as varying spatially – typically employing that prescribed for CMIP6 (Checa-Garcia et al. 2018).   CanAM5 offers two modes for modelling $CO_2$ concentrations - as specified time-evolving concentrations; or as a three-dimensional passive tracer driven by land/ocean surface emissions, prognostically derived through interactive coupling with biogeochemical carbon models in the land and ocean.  For example, CanESM5 can be run with prognostic $CO_2$ in concert with specified anthropogenic fossil fuel emissions to simulate atmospheric $CO_2$ concentration through the historical and future periods. Wetland methane emissions simulated by CLASS-CTEM, in contrast, are purely diagnostic. While these emissions respond to changes in climate and atmospheric $CO_2$ concentration (through changes in vegetation productivity), they do not modify atmospheric $CH_4$ concentration, which are specified."

The authors should ensure that each component model is described in sufficient detail to understand what processes it can simulate and how these models interact together in the ESM. As this is an Earth-system model, I am interested in seeing just how the component models interact. The tables in Appendix A only cover what goes through the coupler, so methane emissions that are discussed on lines 125-130 aren't mentioned for instance - are there any others? Here what do the methane emissions do, is there a simple methane oxidation scheme, does it produce water vapour etc.?

We have expanded the description of CanAM (Section 2.1), and we now believe that each component is described in sufficient detail to understand its complexity. We have added a new section to the manuscript titled "2.6 Treatment of greenhouse gases" to specifically address the question of treatment of methane (and other GHGs).

At line 422 the authors mention "five realisations of CanESM2", and then later discuss a 50 member large ensemble of CanESM2. Is it just that only 5 members were submitted to CMIP5? These are first mentioned on line 844. Is there a reference or more detail that can be provided?

Yes this is correct. Five realizations of CanESM2 were submitted to CMIP5. Later, the 50 member larger ensemble was created by branching those five realizations into ten each in the year 1950 (by perturbing a random seed). To clarify this we have added the text below to the start of Section 6.2, in which the CanESM2 large ensemble data are used:

"Here we make use of the CanESM2 50-member large initial condition ensemble (Kirchmeier-Young et al., 2017; Swart et al., 2018). The 50 realizations in this ensemble

were branched in the year 1950 from the five CanESM2 realizations submitted to CMIP5, and were forced by CMIP5 historical (1950 to 2005) and Representative Concentration Pathway (RCP) 8.5 (2006 to 2100) forcing. ”

**Minor Comments**

Out of interest, why does the numbering system go from CanESM2 to CanESM5?

To clarify this we have added the text to Section 1:

"The leap from version 2 to version 5 was a one-off correction made to reconcile our internal model version labelling with the version label released to the public."

line 33 - I believe that "earth" should be capitalised in this context.

Changed.

line 149 - I'm not sure why "converted into ice" is in quotes here. Is there a reason for this?

The quotes have been deleted. They were used as ice mass is not explicitly tracked, as described, and hence the conversion into ice is implicit.

Section 2.3 - the way numbers and units are represented here are slightly different from other sections. The × symbol is rather large (compare to line 236), and units are given as m2/s rather than m2s−1 as is done elsewhere in the manuscript.

Changed.

line 280 - why are "bilinear" and "conservative" in quotes? I don't think this is necessary here.

Changed.

Section 5.4 - two different ways of doing a ±symbol are used (lines 621 and 642).Given the differences in Section 2.3 (mention above) I suggest that the authors double-check how numbers and symbols are presented for consistency.

Changed.

Table B1 (versioning) - should this be "CanESM.vX.Y.Z"?

Changed.

---

## Author Comment (AC2) · 21 Sep 2019

Review comments in black.

Responses in green.

Review of Swart et al. 2019: The Canadian Earth System Model version 5 (CanESM5).

**General Comments:**

This paper gives a clear and detailed description of the new Canadian Earth System model (CanESM5). The paper nicely describes the main developments relative to earlier Canadian models (e.g. CanESM2) and also provides a good overview of the primary model performance, sensitivities and associated biases, touching on (i) pre-industrial coupled climate stability, (ii) performance over the recent historical past and(iii) first-order sensitivity to increasing CO2 concentrations. It will be a useful reference to users of the model and to climate analysts using CanESM5 CMIP6 results. I recommend the paper be published, conditional on one relatively "major" comment and a number of more "minor" comments being addressed. These are detailed below.

I will structure my comments with some more major ones first followed by a small number of minor errors. I believe that with some small changes to the manuscript the paper would be ready for publication.

We thank the reviewer for their constructive comments.

**Major Comment:**

The paper assesses the performance of CanESM5 over the latter part of the CMIP6historical simulation period, using a number of standard diagnostics where CanESM5 is compared to observations. A number of model biases/shortcomings are identified, but the likely/probable cause of these biases is never discussed. While I recognise it is not straightforward to fully identify the underlying cause of coupled model errors, it would be a useful addition to the paper if, at least for the most important biases(e.g. excessive surface temperature trends post-1980, precipitation and cloud biases, excess heat uptake in the tropical ocean thermocline etc) some discussion of the cause of these biases was provided. This does not have to be exhaustive and can point to a deeper analysis and discussion to come elsewhere, but some pointers to the main causes of key biases would give the paper a bit more "meat". This comment also applies to the significant change (increase) in TCR and ECS in CanESM5 relative to CanESM2, a somewhat greater discussion on what is considered the leading cause(s) for this increase seems appropriate.

We have added to the existing descriptions of possible reasons for the largest biases as follows:

On equatorial thermocline biases (Sec. 5.4) we have added, such that it now reads:

"The pattern of excessive heat accumulation in the thermocline is very similar to the pattern of bias seen in CMIP5 models on average (Flato et al., 2013 their Fig. 9.13). Also similar to CMIP5 models there is a cold bias in the ocean below the thermocline. This suggests that the processes controlling the redistribution of heat between the thermocline and the deep ocean play a role in establishing the vertical structure of these temperature biases. For example, Saenko et al. (2012) find that heat redistribution in ocean models can be sensitive to the vertical structure of diapycnal mixing. "

On the MOC strength (Sec. 5.4), we have added:

"The fairly weak AMOC in CanESM5 is likely associated with excessive sea-ice cover in the Labrador Sea, which inhibits convection."

On precipitation and cloud biases (Sec. 5.2) we have added:

"The overall pattern of precipitation and cloud biases is very similar to that seen across the CMIP5 models (Flato et al., 2013) and are in part related to the well known issues of a double intertropical convergence zone (ITCZ), and an excessive equatorial Pacific cold tongue (Lin, 2007; Li and Xie, 2014)."

On ECS in section 6.1 we have added to the existing description, such that it now reads:

"A detailed explanation of the reasons behind the increased ECS in CanESM5 over CanESM2 is beyond the scope of this paper. However, the effective radiative forcing (Forster et al, 2016) in CanESM5 due to abrupt quadrupling of $CO_2$ is very similar to that in CanESM2, suggesting that changes in feedbacks rather than forcings are the source of the higher ECS. Indications are that the increase in ECS is associated with cloud and surface albedo feedbacks, with sea-ice likely playing an important role in the latter effect. The cloud albedo feedback is found to be sensitive to parameter settings in the cloud microphysics scheme. A more detailed examination of the changes in ECS due to cloud microphysics will be provided in a companion paper in this special issue (Cole et al, 2019). "

On historical warming rates, we discuss the possible causes in Sec. 6.2.1 in which we raise four possibilities, we rule out internal variability, and we suggest that the increase in historical warming is most likely driven by the increases in ECS.

Overall we have tried throughout Sections 5 and 6 to provide concise descriptions of our best understanding of the causes of biases. However as the reviewer notes, it is often extremely difficult to identify the cause of coupled model biases. Indeed, many of the biases noted are common across many models, and their causes remain outstanding questions within the broader community. We feel that it is beyond the scope of this particular paper to explore the causes of biases in further detail. The primary goal of this paper is to provide an overview of the model code and a basic description of model performance to set the stage for further usage and analysis. We note that the remainder of the special issue is available for a more detailed analysis of biases (and we intend to use it as such).

**Minor comments:**

Minor comments are listed with reference to the line numbers in the paper. Where I think some additional explanation would improve the paper I outline this within each point.

1. L50: "CanESM5 represents a major update to CanESM2": The reader is left wondering what happened to CanESM3 and 4 ?? Is there some reason naming jumped from "2" to "5"?

To clarify this we have added the text to Section 1:

"The leap from version 2 to version 5 was a one-off correction made to reconcile our internal model version labelling with the version label released to the public."

2. L86-L87: "the emission of mineral dust and DMS was improved...". It would be useful to know the degree to which dust, DMS and other natural (aerosol) emissions are fully prognostic in the model, allowing potential future feedbacks to be simulated, versus these using observation-based (and therefore time-invariant) prescribed input fields.

Section 2.1 has been expanded to clarify the treatment of aerosols:

"The aerosols are parameterized using a prognostic scheme for bulk concentrations of natural and anthropogenic aerosols, including sulfate, black and organic carbon, sea salt, and mineral dust; parameterizations for emissions, transport, gas-phase and

aqueous-phase chemistry, and dry and wet deposition account for interactions with simulated meteorology."

3. L101-L102: "The introduction of dynamic wetlands and their methane emissions is a new biogeochemical process...". Is methane a prognostic atmospheric variable inCanESM5? e.g. how do the wetland methane emissions influence the model radiation? There is no mention of interactive chemistry in CanESM5, I therefore assume methane is not a prognostic variable and that there are no internal (model) feedbacks between wetland methane emissions and climate. How methane (and other non-CO2 radiatively important gases e.g. ozone) are treated in CanESM5 needs to be more clearly explained.

We have added a new section to the manuscript (2.6) subtitled "The treatment of greenhouse gases", to address this question, which explains that methane is not prognostic (it is specified), as is ozone. For completeness, we have also added comparison of wetland extent simulated by CanESM5 with observation-based estimates and the temporal evolution of annual maximum wetland extent and wetland methane emissions over the historical period in a new section in the appendix.

4. L121: Why does CLASS use 4 PFTs but CTEM use 9? And is vegetation type/amount dynamically predicted in CanESM5 or (externally) prescribed? Please state what is done for the latter and explain the rationale for the former.

The following text is added/modified to Section 2.2 to address this question:

"The reason for separation of PFTs for CTEM is the additional distinction that biogeochemical processes require. For example, the distinction between deciduous and evergreen versions of needleleaf trees is needed to simulate leaf phenology prognostically. Once leaf area index has been dynamically determined by CTEM all CLASS needs to know is that this PFT is needleleaf tree since the physics calculations do not require information about underlying deciduous or evergreen nature of leaves. Similarly, the $C_3$ and $C_4$ photosynthetic pathways of crops and grasses determine how they photosynthesize thus affecting the calculated canopy resistance. However, once canopy resistance is known CLASS does not need to know the underlying distinction between $C_3$ and $C_4$ crops and grasses to use this canopy resistance in its energy and water balance calculations.

While, the modelled structural vegetation attributes respond to changes in climate and atmospheric $CO_2$ concentration, the fractional coverage of CTEM's nine PFTs is

specified. A land cover data set is generated based on a potential vegetation cover for 1850 upon which the 1850 crop cover is superimposed. From 1850 onwards, as the fractional area of $C_3$ and $C_4$ crops changes the fractional coverages of the other non-crop PFTs are adjusted linearly in proportion to their existing coverage, as described in Arora and Boer (2010). The increase in crop area over the historical period is based on LUH2 v2h product (http://luh.umd.edu/data.shtml) of the land use harmonization (LUH) effort produced for CMIP6 (Hurtt et al., 2011)."

5. L304-306: It sound as though the primary tuning for the coupled pre-industrial model simulation was global mean surface temperature. Is this correct? It would seem to me that (a zero) global mean net top of atmosphere radiation would be a more appropriate target. The resulting net TOA balance looks to fairly close to zero (0.11Wm-2, figure 5a). Was the TOA net radiation budget also used as a tuning target?

Yes we absolutely agree. We implicitly meant achieving a reasonable global mean surface temperature with a TOA balance of (near) zero - but we have now made this explicit as follows by adding to this sentence:

"while maintaining a net TOA radiative balance as close to 0 W m$^{-2}$ as possible"

6. L318-319: "The final adjustment was to the carbon uptake over land......achieved via the parameter which controls the strength of the CO2 fertilization effect". Does this indicate some structural problems with the parameterization of CO2 fertilization? I ask because in L813-815 there is mention of; "significantly larger uptake of CO2 by the land biosphere in CanESM5 relative to CanESM2 in the 1pctCO2 experiment". Is this change in land uptake influenced by the tuning applied to the historical period CO2 fertilization term? Allied to this does CanESM5 include any representation of nutrient limitation of the land CO2 fertilization term?

The following text is added to section 2.2 to address this comment:

"Both CanESM5, and its predecessor CanESM2, do not include nutrient limitation of photosynthesis on land since terrestrial nitrogen cycle is not represented. However, both models include a representation of terrestrial photosynthesis downregulation based on Arora et al. (2009) who used results from plants grown in ambient and elevated $CO_2$ environments to emulate the effect of nutrient constraints. The tunable parameter determining the strength of this downregulation, and therefore the strength of the $CO_2$ fertilization effect, is higher in CanESM5 than in CanESM2 resulting in higher land carbon uptake in CanESM5. The tuning of this downregulation parameter value, used in CanESM5, is explained in Arora and Scinocca (2016) who evaluate several aspects of modelled historical carbon cycle against observation-based estimates. A land nitrogen cycle model for CTEM is currently being developed which will make the photosynthesis downregulation parameterization obsolete in future versions of the model."

We also add in Section 6.1:

"The reduction in TCRE occurs despite the fact that CanESM5 has a larger temperature response (TCR) than CanESM2 due to significantly larger uptake of $CO_2$ by the land biosphere in CanESM5 relative to CanESM2 in the 1pctCO2 experiment. As mentioned in Section 2.2, this is due to higher strength of the $CO_2$ fertilization effect in CanESM5 relative to CanESM2. As shown in Arora and Scinocca (2016) this leads to land carbon uptake in the 1pctCO2 simulation that is higher than in all CMIP5 models compared in Arora et al. (2013)."

7. L320: "Critically, no tuning was undertaken on the climate system response to forcing". This is not strictly true as the authors say that they tuned the model to improve aspects of the historical simulation, which does include perturbed CO2 forcing.

The tuning based on the historical simulation was for land carbon uptake. In specified CO2 simulations (i.e. most of CMIP6, including the abrupt-4xCO2 simulation), land carbon uptake is purely diagnostic and does not influence climate sensitivity, which is what we are referencing here. To make this more accurate/explicit, we have changed this line to:

 "no tuning of climate sensitivity was undertaken".

8. Figure 4: This figure appears on page 15 but is not discussed until much later. In itself that is okay, but then it might be worth adding a statement where the figure is first shown, saying why this figure is sufficiently important to be shown at the start of this section and that further discussion of the figure results appear in sections xx or yy.

We have added to the text :"Fig. 4 shows the global mean surface temperature for several of the key CMIP6 experiments, which can be used to infer important properties of the model, as discussed further in Sections 4 to 6"

9. Figure 5 (caption). Some of the figures are explicitly listed as being "global" or "global mean" while others are not. Those not classified as "global" or "global mean", if they are not, for what spatial region are they representative? If they are all global quantities then make this clear in the figure caption.

The Figure caption has been clarified to indicate that all quantities are global means.

10. Figure 6 (caption): "Variables are labelled according to the names in the CMIP6 data request" As not all readers will be CMIP6 experts, it could be useful to list what this shorthand list means in real word descriptions, possibly in a table in an appendix.

We have inserted table F3, which includes the CMIP6 names and (real world descriptive) long names of each variable used, and we reference this new table in the caption to Figure 6.

11. L520-521: Some of the precipitation biases, while large in percentage terms, are actually very small in absolute terms (e.g. the subtropical East Pacific). Would it be better to plot absolute precipitation biases? as done for temperature and SLP.

We have converted this figure to use absolute biases

12. For Figure 7 (and possibly 8). I suggest showing DJF and JJA seasonal means rather than annual means. This will help focus on biases and reduce the risk of error cancellation within the annual mean, e.g. winter vs summer surface temperature biases, better focus on summer vs winter monsoon precipitation biases etc.

We have converted this figure to show DJF and JJA seasonal means (new Figs. 5, 6, 7).

13. Figure 8 and L530: It would be useful to plot TOA solar and longwave radiation fields and biases (e.g. vs CERES EBAF) alongside the cloud plots, given that the number 1 consequence of cloud errors are in their impact on radiation.

Figure 8 has been expanded to show seasonal means, as requested above. We have elected to leave more detailed comparisons (vs CERES) to the upcoming documenting papers that are specifically focused on CanAM.

14. L589-591: The wording in these lines does not make sense.

Thanks for pointing this out typo. We have adjusted the text to read as follows: "The observations-based temperature and precipitation data used in these plots are from CRU-JRA reanalysis data that were used to drive terrestrial ecosystem models in the TRENDY Intercomparison for the 2018 Global Carbon Budget (Le Quéré et al., 2018)."

15. Figure 13: (caption) "The shading presents the corresponding inter-quartile range": What range are we referring to here? For the model results is the range across ensemble members? What is the range with respect to the observations?

The shading presents the inter-quartile range that results from the combined effect of inter-annual variability and longitudinal variability. Model results show the ensemble member r1i1p1f1 only. We have adjusted the Figure caption as follows: "Figure 13: Zonal mean values of (a) GPP, (b) HFLS, and (c) HFSS for CanESM5 (r1i1p1f1) (black) and reference data (red) from Jung et al. (2009). The shading presents the corresponding inter-quartile range that results from inter-annual variability as well as longitudinal variability for the period 1982 to 2008."

16. Figure 14: What occurrence of surface temperature and precipitation is plotted? Daily mean values, monthly means? Please state what is plotted.

The values present monthly mean values averaged over the period 1982 to 2008. We have adjusted the Figure caption as follows: "Functional response of GPP to (a) near-surface air temperature and (b) surface precipitation for CanESM5 (r1i1p1f1) and reference data from Jung et al. (2009). Values present monthly mean values averaged over the period 1982 to 2008 and the shading shows the corresponding standard deviation."

17. L615-616: "Negative annual mean SSS biases occur under the region associated with excessive March sea-ice..." This seems counter to what I would expect. Generally with sea-ice formation there is brine rejection into the surface water, hence with excessive sea ice I would expect excessive brine rejection and a positive bias in SSS.

We have deleted "under the region associated with excessive March sea-ice". Our statement was based on the visual coherence between the patterns of sea-ice bias, with SSS (and SST). Mechanistically, negative SSS biases could be expected if sea-ice formed elsewhere is being advected into this region where it subsequently melts.

18. L676-678: Please provide references to PIOMAS and GIOMAS.

We inserted in-text references to Zhang and Rothrock (2003) and Schweiger et al., 2011.  These are also provided in Table F2.

19. Figure 20: The excessive model sea-ice in the Labrador sea, does this have an impact on deep water formation and potentially help explain the weak (~12Sv) AMOC in CanESM5?

Yes, we expect that it does. The AMOC is stronger in ocean only simulations which do not have this ice cover over the Labrador sea, although, there are also other differences between the coupled and ocean only model (no coupled feedbacks), so this is not definitive.

20. Figure 20 needs a clearer figure caption and explanation. It is not clear whether the plot shows ocean to atmosphere or atmosphere to ocean C flux. I am guessing positive values in the North Atlantic indicate ingassing (e.g. atmosphere to ocean flux of C).The implied direction of the flux with respect to positive and negative values needs to be made clearer. Also figure b looks like the difference in flux CanESM5 minus Landschutzer, the figure caption suggests it is an absolute value of Landschutzer.

We assume here that the reviewer is referring to Fig. 22, which shows the atmosphere-ocean $CO_2$ flux (Fig 20 is sea-ice). We have inserted a clarification on the flux direction into the Fig. 22 caption:

"The flux is positive down (into the ocean)"

and we have clarified that panel b) is the bias, not absolute value of Landschutzer:

"(b)  the bias relative to Landschutzer (2009)"

21. L773-774: Why use this definition of the Northern Annular Mode? Is this a standard circulation measure? Would it not be better to plot the NAO for the Northern hemisphere?

We use the standard definition of the Northern Annular Mode, the first EOF of sea level pressure north of 20 degrees north, for example as given in Thompson and Wallace (1999) Annular Modes in the Extratropical Circulation. Part I: Month-to-Month Variability, J. Climate, 13, and also as at

 https://climatedataguide.ucar.edu/climate-data/hurrell-wintertime-slp-based-northern-annular-mode-nam-index

22. Figure 24 (caption): "The colour scale is arbitrary" What does this mean? Presumably the scales are the same across the 4 figures?

The scale is the same across figures. We have removed this statement from the caption.

23. L813-816: The TCR and ECS both increase (by~16% and 50%) in CanESM5relative to CanESM, yet the TCRE decreases by~16%. This is unusual. It is

explained as being due to a "significantly larger uptake of CO2 by the land biosphere in CanESM5relative to CanESM2". Given the stated importance of TCRE for policymakers, are the authors confident that the increased CO2 uptake is due to realistic and understood reasons?

The land carbon uptake in Earth system models remains highly uncertain as has been noted in Friedlingstein et al. (2006, https://journals.ametsoc.org/doi/10.1175/JCLI3800.1) and Arora et al. (2013, (https://journals.ametsoc.org/doi/full/10.1175/JCLI-D-12-00494.1).  As mentioned above, in reply to comment # 6, the tunable parameter determining the strength of this downregulation, and therefore the strength of the CO2 fertilization effect, is higher in CanESM5 than in CanESM2 resulting in higher land carbon uptake in CanESM5. The tuning of this downregulation parameter value, used in CanESM5, is explained in Arora and Scinocca (2016, https://www.geosci-model-dev.net/9/2357/2016/) and uses the land carbon uptake and other aspects of the carbon cycle over the historical period to obtain the best parameter value. Yet, of course, realistic model performance over the historical period is no guarantee of realistic model performance for future periods. Note that modelled historical land carbon uptake in CanESM5 is lower than observation-based estimates from Le Quere et al. (2018) yet its land carbon uptake in the 1pctCO2 simulation is highest amongst all CMIP6 models (manuscript currently in preparation by Vivek Arora). This requires a thorough investigation of CanESM5 results to find reasons behind this apparent conundrum. We suspect that forcing differences between the historical and 1pctCO2 experiments (aerosols in particular), influence rainfall changes, and thereby land carbon uptake. So yes, indeed, the reasons for lower TCRE despite higher TCR and ECS are well understood.

The text in section 6.1 which discusses TCRE is modified as follows:

"The reduction in TCRE occurs despite the fact that CanESM5 has a larger temperature response (TCR) than CanESM2 due to significantly larger uptake of $CO_2$ by the land biosphere in CanESM5 relative to CanESM2 in the 1pctCO2 experiment. As mentioned in Section 2.2, this is due to higher strength of the $CO_2$ fertilization effect in CanESM5 relative to CanESM2. As shown in Arora and Scinocca (2016) this leads to land carbon uptake in the 1pctCO2 simulation that is higher than in all CMIP5 models compared in Arora et al. (2013)."

and in Section 6.2.3:

"In contrast, the simulated mean atmosphere-land $CO_2$ fluxes (Fig. 28c) for the decades of 1960s through to 2000s are lower than their observation-based estimates from Le Quéré et al. (2018). This is despite the fact that the land carbon uptake in the 1pctCO2 simulation for CanESM5 is highest amongst all CMIP5 models reported in Arora et al. (2013). The reason for this conundrum is a topic for future investigation, but might relate to differences in forcing (aerosols) in the historical and 1pctCO2 experiments."

24. L821-825: "ECS IS 5.7K": Actually, in figure 4 the global mean surface temperature does not look to have sufficiently stabilized by year 140 to derive an ECS value (e.g.it increase by~0.75C over years 90 to 150). Have the authors tried plotting the so-called Gregory linear regression of nett TOA radiation versus surface temperature to see whether there is a single or multiple linear relationships between these two, or whether a non-linear behaviour is evident?

The ECS value was calculated using the Gregory linear regression approach, not by looking at Figure 4. In the text we state "Given the long equilibration time of the climate system, it is common to estimate ECS from the relationship between surface temperature change and radiative forcing, over the course of the first 140 years of the abrupt-4xCO2 simulation (Gregory et al., 2004).". To clarify that the number we report was calculated in this way, we have modified the sentence referenced by the reviewer to read:

"Here the ECS is calculated using the Gregory (2004) regression method, after removing linear drift from the piControl following Forster et al. (2013)".

The plot of TOA radiation vs surface temperature in CanESM2 and CanESM5 over the first 140 years of the abrupt-4xCO2 experiment, together with the regression lines is included below for reference. We note a slight refinement to the CanESM5 ECS value (5.6) in the revision, relative to the original (5.7), owing to the removal of the linear trend from the piControl.

[Figure]

25. Figure 26c: It is difficult to see any differences. This might be better plotted as a model to observation difference or ratio.

We show multiple observations, so a simple difference or ratio is not as intuitive. We agree that differences in low latitudes are harder to see (overlapping lines) as they are small. However where differences are large, such as over the Arctic, they can easily be seen. We choose to leave this figure as it is.

---

## Author Comment (AC3) · 21 Sep 2019

Review comments in black.

Responses in green.

Review of The Canadian Earth System Model version 5 (CanESM5.0.3) by Neil et al. for GMD.

The paper describes the last version of CanESM. The main goal is to provide a reference for people who will analyse CMIP6 model outputs. Like all model description paper, there is a conflict between the need to be as comprehensive as possible, while keeping a reasonable paper length. In this point of view, the paper achieves a good and relevant compromise. It includes a short description of the model components, with all references for the reader who wants to go further in details. It contains a classical choice of model diagnostic to evaluate the model climate. It contains also informations, of the model sensitivity to standardised scenarios of CO2 increase. This content nicely matches the main objective of the paper.

I really appreciated the honesty of the authors, particularly when they described the consequence of the bug in p1 version, corrected in the p2 version. The paper also includes information about model quality control and performance. This is not of interest for CMIP6 data users. But model developers will find useful information on the way other teams work. Information that is hardly presented elsewhere. The general organisation of the paper is good. As non-native English speaker, I won't comment the quality of the syntax. I found very few typos, and thanks the authors for their careful proof reading. The main weakness of the paper is that the model is assessed in comparison with data and with the previous version of the model. There is no comparison with the CMIP5 models. This is not a major concern, as the model outputs will be used during CMIP6, and CamESM will be compared to CMIP5 and CMIP6 model in the next months by other authors. To sum up, this is a very good paper that perfectly fits the reason why GMD has been created for. I think it can be published after a few technical corrections.

We thank the reviewer for their constructive comments.

**Major concern**

Line 351. I do not understand how "version control, run isolation, strict checking and logging" can insure that the climate is reproducible. I agree that up to now, nobody has observed that lack of bit identical reproducibility in an ESM can drive to a different climate. But we have also theories of deterministic chaos showing that this is possible.

The system described enables us to ensure that we can re-run precisely the same code, in the same way as an original run. On a given machine, the results are bit identical. In some cases, moving across machines (e.g. our current migration from Cray XC40 to Cray XC50) also allows us to maintain bit identity. Hence these runs are precisely numerically reproducible.

Migrations to a different architecture or compiler might result in a bit pattern change. In this case our expectation (and experience) is that the climate will remain the same, but the realization of internal variability will be different. In this sense the run is reproducible in that we are interested in climate not weather. We accept that it is theoretically possible that mulitple-equilibria exist within the model in general. However, within fully coupled modelling this has essentially never been observed for a modern-day like climate, as the reviewer states. The only mulitple-equilibria we know of in complex GCMs involve radical mean state / forcing changes such as under extensive glaciation. There is no evidence we know of that bit-scale changes can lead to a different climate state.

Indeed, we note that different initial condition realizations of the model - with a similar type of small perturbation - result in the same climate. The concept that there are infinitely many possible realizations with the same climate (rather than different equilibrium climates) is also widely employed in coupled modelling and international exercises such as CMIP, which make extensive use of initial condition ensembles. Thousands of such initial condition simulations have been conducted, without the appearance of different equilibrium climates, as far as we know. It is impossible to ever prove that bit-induced multiple equilibria do not exist - but the extensive number of previous simulations are evidence that bit-induced multiple equilibria are exceedingly

unlikely. For these reasons we are confident that the CanESM5 climate is robust to bit-pattern changes.

**Minor concerns**

Line 125 and following. Is there some specific representation of the urbanised areas ?

There is no explicit model for urban areas. Urban areas are represented/parameterized by higher albedo for visible light and lower albedo for near infrared radiation than for natural vegetation. The roughness length over urban areas is higher than that for crops and grasslands but lower than for trees.

Line 151 and following. The melt water of the glaciers goes to the runoff scheme. How did you design a 'river' routing scheme for the ice sheets ? (from slopes ?)

There is not a dedicated runoff scheme for ice sheets. When the ice melts, the liquid meltwater is treated like runoff and the river routing scheme carries it down stream to the nearest ocean grid cell. This is do-able since river flow directions are specified over glacial cells as well based on their topography. Please see Figure 1 of Arora and Boer (1999).

Line 162. For the NEMO TKE scheme, a better reference than Gaspar et al. 1990 could be Blanke, B. and Delecluse, P.: Variability of the Tropical Atlantic Ocean simulated by a general circulation model with two mixed layer physics, Journal of Physical Oceanography, 23, 1363–1388, 1993.

This reference has been inserted.

Line 218. What is the computing coast of CMOC and CanOE compared to NEMO dynamics and to LIM2 ? It is significant or not ?

Yes it is significant. Turning on CMOC reduces model throughput by a factor of 2, relative to the physical model. Turning on CanOE reduces throughput by a factor of 4 relative to the physical model.

Line 777. Bentson et al. 2013 is not in the bibliography. Probably a typo, for Bentsen.

Corrected.

Line 809. Mathews et al. 2009 is not in the bibliography. Probably a typo, for Matthews.

Corrected.

Line 840. What is "Global Mean Screen Temperature" ? GMST generally stands for Global Mean Surface Temperature.

Changed "Screen" to "Surface".

Line 1335. Year of the paper is embedded in the URL of the DOI.

Fixed.

---

## Author Comment (AC4) · 21 Sep 2019

Comment: 12. For Figure 7 (and possibly 8). I suggest showing DJF and JJA seasonal means rather than annual means. This will help focus on biases and reduce the risk of error cancellation within the annual mean, e.g. winter vs summer surface temperature biases, better focus on summer vs winter monsoon precipitation biases etc.

Reponse: Figures 7 and 8 have been split and updated to show seasons DJF and JJA, as suggested, and shown attached. The new figures will be Figs. 5 through 8.

[Figure]

[Figure]

**Fig. 1.** Update of fig. 7 annual mean tas to DJF and JJA

[Figure]

**Fig. 2.** Update of fig. 7 annual mean pr to DJF and JJA

[Figure]

**Fig. 3.** Update of fig. 7 annual mean psl to DJF and JJA

[Figure]

**Fig. 4.** Update of fig. 8 annual mean clt to DJF and JJA